# 19[th] century glacier retreat in the Alps preceded the emergence of industrial black carbon deposition on high-alpine glaciers, casting doubt on a leading role for anthropogenic BC emissions in terminating the Little Ice Age

Michael Sigl[1,2], Nerilie J. Abram[3], Jacopo Gabrieli[4], Theo M. Jenk[1,2], Dimitri Osmont[1,2,5], Margit Schwikowski[1,2,5]

[1]Laboratory of Environmental Chemistry, Paul Scherrer Institut, 5232 Villigen, Switzerland
[2]Oeschger Centre for Climate Change Research, University of Bern, 3012 Bern, Switzerland
[3]Research School of Earth Sciences and the ARC Centre of Excellence for Climate System Science, The Australian National University, Canberra 2601 ACT, Australia
[4]University Ca'Foscari, University of Venice, 30123 Venezia, Italy
[5]Department of Chemistry and Biochemistry, University of Bern, 3012 Bern, Switzerland

*Correspondence to*: Michael Sigl (michael.sigl@psi.ch)

**Abstract.** Light absorbing aerosols in the atmosphere and cryosphere play an important role in the climate system. Their presence in ambient air and snow changes radiative properties of these media, thus contributing to increased atmospheric warming and snowmelt. High spatio-temporal variability of aerosol concentrations and a shortage of long-term observations contribute to large uncertainties in properly assigning the climate effects of aerosols through time.

Starting around 1860 AD, many glaciers in the European Alps began to retreat from their maximum mid-19[th] century terminus positions, thereby visualizing the end of the Little Ice Age in Europe. Radiative forcing by increasing deposition of industrial black carbon to snow has been suggested as the main driver of the abrupt glacier retreats in the Alps. The basis for this hypothesis were model simulations using elemental carbon concentrations at low temporal resolution from two ice cores in the Alps.

Here we present sub-annually resolved, well-replicated concentration records of refractory black carbon (rBC; using soot photometry) as well as distinctive tracers for mineral dust, biomass burning and industrial pollution from the Colle Gnifetti ice core in the Alps from 1741-2015 AD. These records allow precise assessment of a potential relation between the timing of observed acceleration of glacier melt in the mid-19[th] century with an increase of rBC deposition on the glacier caused by the industrialization of Western Europe. Our study reveals that in 1875 AD, the time when rBC ice-core concentrations

started to significantly increase, the majority of Alpine glaciers had already experienced more than 80% of their total 19[th] century length reduction. Industrial BC emissions can, therefore, not been considered as the primary forcing for the initial rapid deglaciation at the end of the Little Ice Age in the Alps. BC records from the Alps and Greenland also reveal the limitations of bottom-up emission inventories to represent a realistic evolution of anthropogenic BC emissions since

preindustrial times. For accurate quantification of past European BC emissions, however, the spatial network and the sampling density of high-alpine ice cores needs to be expanded to balance potential biasing effects arising from transport, deposition and snow conservation in individual ice-core records.

## 1 Introduction

The role of aerosols in climate forcing (defined as perturbation of the Earth's energy balance relative to the preindustrial) is significant but poorly understood (Charlson et al., 1992). Aerosol emissions and their atmospheric burden vary in time and from region to region; some aerosols cause cooling while even co-emitted species can lead to simultaneous warming. This results in large uncertainties of the ascribed radiative forcing terms to short-lived aerosols in contrast to greenhouse gas forcing (Bond et al., 2013; Dubovik et al., 2002).

Black carbon (BC) has a unique and important role in the climate system because it absorbs –even at very low concentrations solar radiation, influences cloud formation, and enhances the melting of snow and ice via albedo feedbacks (Flanner et al., 2007; Hansen and Nazarenko, 2004). BC is defined as an incomplete combustion product from natural biomass burning (e.g., forest fires) or anthropogenic biofuel and fossil-fuel burning. It is insoluble, refractory, strongly absorbs visible light, and forms aggregates of small carbon spherules. Per unit mass, BC has the highest light absorption of all abundant aerosols

in the atmosphere (Bond et al., 2013). Given that carbonaceous aerosols in the atmosphere present a continuum of varying physical and chemical properties, their quantification is strongly related to the analytical method used. A wide range of terminologies has developed in the scientific community to characterize BC and related carbonaceous aerosols, and we follow the terminology recommendations recently put in place (Petzold et al., 2013). Refractory black carbon (rBC) will be used instead of black carbon for reporting concentrations derived from our laser-based incandescence method, while the

general term black carbon (BC) is used for a qualitative description when referring to light-absorbing carbonaceous substances in atmospheric aerosol. If analyzed with a thermal optical method, BC is also referred to as elemental carbon (EC) (Currie et al., 2002).

While natural sources such as forest fires dominated the global BC burden in the preindustrial atmosphere, current emissions are largely driven by industrial, energy related sources (Bond et al., 2013). The modern burden is highest in heavily

industrialized and populated regions including China, India, and Europe (Fig. 1). Trends in BC emissions estimated from bottom-up approaches (i.e., from fuel consumption data) suggest large changes during the industrial era (Bond et al., 2007; Lamarque et al., 2010), which were recently largely confirmed by continuous measurements of BC in Greenland ice cores (Bauer et al., 2013; Koch et al., 2011; Lee et al., 2013a; McConnell et al., 2007). However, multiple source regions contribute with varying degree to the BC deposition over Greenland, hampering attribution of the observed trends to

individual emission source areas (Hirdman et al., 2010; Liu et al., 2011).

Together with mineral dust and other absorbing organic aerosols, BC deposited on snow and ice can lead to increased melt rates and changes in melt onset due to changes of the surface albedo with these effects being further enhanced by subsequent

snow albedo feedbacks such as an increase in the water content and surface accumulation of impurities (Flanner et al., 2009; Hansen and Nazarenko, 2004). The best estimate for industrial era global forcing is +0.13 W m$^{-2}$, but values for regions with seasonal snow cover (e.g., Arctic, European Alps, Tibetan Plateau) are much higher (Bond et al., 2013). Industrial BC deposition has been suggested to be responsible for observed Arctic warming in the 1940s and recent years

(Flanner, 2013; Flanner et al., 2009; Quinn et al., 2008) but recent surface albedo decreases (i.e. darkening) of the Greenland ice sheet occurred in the face of a widespread decrease in BC deposition based on multiple ice cores (Keegan et al., 2014; McConnell et al., 2007) suggesting a small role for light absorbing impurities in causing these changes (Polashenski et al., 2015). In the Himalaya the combined increased deposition of mineral dust and industrial black carbon was suggested to play a role in the observed glacier retreat during the past decades (Flanner and Zender, 2005; Kaspari et al., 2011; Lau et al.,

2010; Lee et al., 2013b). In contrast to climate effects from direct radiative forcing (Bond and Sun, 2005; Penner et al., 1998) and cloud effects (Haywood and Boucher, 2000; Lohmann and Feichter, 2005) that are short lived and effective only during the brief atmospheric lifetime of the aerosols (days to a week), BC-induced changes in the snow cover persist for longer periods of time ranging from weeks-to-months. They are most pronounced during the spring and summer, when insolation and seasonal snowmelt reach a maximum (Flanner et al., 2009).

To quantify trends and magnitudes of climate forcing from BC in the atmosphere (direct and indirect effect) and cryosphere (snow-albedo-effect) climate-model simulations are widely used (Bond et al., 2013; Lamarque et al., 2013; Shindell et al., 2013). These rely strongly on energy-consumption based estimates of BC emissions which are highly uncertain (see Figure 8 in Bond et al., (2007)) and therefore need to be evaluated against independent ice-core based observations (Bauer et al., 2013; Lee et al., 2013a). Those comparisons allow identification of mismatches and can subsequently help to improve

parameterization and model performance (Lamarque et al., 2013).

Mountain glaciers are retreating worldwide and are projected to further shrink with the expected increase in global surface temperatures due to increasing greenhouse gas concentrations (Mernild et al., 2013; Oerlemans, 2005; Zemp et al., 2006). While the currently observed mass loss is global in scale and attributed to anthropogenic greenhouse gas emissions, the onset of melting during the 19$^{th}$ century was asynchronous for many mountain regions (e.g., between Scandinavia and the Alps)

(Imhof et al., 2012; Larsen et al., 2013). Observations place the start of the retreat in the western Alps from 1860 to 1865 after glaciers reached their maximum extent around 1850-1855 (Nussbaumer and Zumbühl, 2012; Zumbühl et al., 2008). The retreat was rapid and synchronous among different documented glaciers. By 1880, glacier tongues had retreated by several hundred meters in length (Nussbaumer and Zumbühl, 2012). Using early instrumental temperature and precipitation data, a combination of high spring temperatures and reduced autumn precipitation was suggested as the main drivers of the

observed glacier retreat (Steiner et al., 2008; Zumbühl et al., 2008). In an alternative hypothesis, early industrial BC deposited on snow and ice of Alpine glaciers was held responsible for the rapid melting, involving a snow-albedo feedback (Painter et al., 2013). This hypothesis built on model simulations to estimate snow albedo forcing from two ice-core based reconstructions of BC (i.e. EC) from Fiescherhorn glacier (Jenk et al., 2006) and Colle Gnifetti (Thevenon et al., 2009). Starting in the 1870s, both records show an initial 2-3-fold increase of BC concentrations rising from a mostly natural

background of 9 ng g$^{-1}$ to more than 20 ng g$^{-1}$, before the highest values (37 ng g$^{-1}$) were reached during the early 20[th] century. Transient changes in external natural and anthropogenic climate forcing during the emergence of the industrialization in Europe potentially intertwined with non-linear feedback mechanism (such as an effect of rBC on albedo) require comprehensive modelling efforts to isolate the often-complex relationships between glacier fluctuations and meteorological forcing, in order to demonstrate the mechanisms responsible for glacier retreat in the second half of the 19[th] Century (e.g., Lüthi 2014; Zekollari 2017, Goosse et al., 2018). Underpinning such detection and attribution efforts, accurate and precise delineation of external forcing, potential feedbacks and cryosphere changes is critically important.

The snow-albedo feedback hypothesis formulated by Painter et al. (2013) was a first effort to attempt this but was naturally limited by the BC data available at the time and thus compiled from very first applications using different analytical methods, performed in low temporal resolution and partly published on preliminary timescales only. Some of these methods turned out to be unable to deliver reproducible results and thus were not applied in any future investigation since. The obvious choice was to use glacier length curves with comparable low temporal resolution as the available BC records at the time did not allow resolving the precise timing of the BC increases. Clearly, their conclusions were also restricted by not considering contributions from other absorbing aerosols present in the snow (e.g., Saharan dust; brown carbon; macroscopic charcoal or pollen) which however are admittedly not well constrained (Brugger et al., 2018; Oerlemans et al., 2009; Skiles et al., 2012; Sun et al., 2017) and therefore might not have been incorporated into the radiative transfer model.

Here we set out to re-evaluate the timing of industrial BC deposition in ice cores from the Alps which is key information to assess the hypothesis of a strong role of industrial BC in forcing of the "End of the Little Ice Age in the Alps" (Painter et al., 2013). Years later, we are now in the fortunate position allowing us to do this by using a new, more accurately dated record of light absorbing aerosols in much higher resolution (i.e., BC and mineral dust), in combination with a suite of other distinctive tracers of anthropogenic pollution and in tandem with the most highly resolved history of glacier length changes of four glaciers in the Western Alps currently available (Nussbaumer and Zumbühl, 2012). We determine the time when industrial BC emerged from the preindustrial background relative to the observational history of terminus position of European glaciers and evaluate to which extent these results were consistent with the hypothesis put out for future investigations by Painter et al. (2013) of a strong ice-albedo feedback starting in the 1850s through increased ablation rates at the glaciers surface.

## 2. Data and Methods

### 2.1 Ice core drilling site

Two 82 m long ice cores (CG03A, CG03B) from the European Alps, which are surrounded by highly-industrialized countries (e.g., Germany, Italy, France) representing some of the main emitters of 19[th] to 21[st] century fossil-fuel industrial BC (Bond et al., 2007) were drilled in 2003 a meter apart on Colle Gnifetti (Monte Rosa, 4450 m a.s.l., 45°55'55"N; 07°52'34"E, (Jenk et al., 2009)) (Fig. 1). The drill site location on a saddle minimizes effects of lateral ice flow, but leads to

seasonally weighted atmospheric signals, due to preferential wind erosion of winter snow (Bohleber et al., 2013; Häberli et al., 1983; Oeschger, 1977; Schwikowski et al., 1999a). Low annual net accumulation rates (0.45 m water equivalent yr$^{-1}$) give access to an old age of the ice and allow to retrieve long-term proxy records covering most of the Holocene (Jenk et al., 2009; Konrad et al., 2013; Sigl et al., 2009). During the summer months the drill site is frequently situated above the planetary boundary layer. However, the site clearly records the signal of natural and anthropogenic emissions from sources located at lower altitudes as indicated by the diurnal and annual cycle of aerosol deposition resulting from convective transport monitored in-situ at Colle Gnifetti and Jungfraujoch over many years by remote sensing and high-resolution aerosol measurements (Lugauer et al., 1998; Nyeki et al., 2000). The presence of these annual cycles is the basis for dating of these ice cores by incremental layer-counting reaching as far back as 1,000 years (Bohleber et al., 2018). On inter-annual timescales, the summer-biased and irregular snow deposition at Colle Gnifetti contributes to the observed variability of the proxy records, with occasionally preserved winter snowfall (e.g., in ~1890 AD) typically having low impurity concentrations (Wagenbach et al., 2012). These effects are minimized, however, at a (multi-) decadal resolution, at which these proxy records reflect remarkably well changes in the source strengths of the emissions and the resulting atmospheric burden of aerosols (Engardt et al., 2017; Fagerli et al., 2007; Gabrieli et al., 2010; Schwikowski et al., 1999b). Additional firn/ice cores (CG08, CG15) were obtained from the same site in 2008 (Kirchgeorg et al., 2013) and in September 2015, respectively, to update the long-term record to the most recent past.

The Colle Gnifetti site has produced a number of impurity and pollution records, highlighting the strong impact of human activity on the atmospheric composition over Central Europe during the last decades and centuries, including records of sulphate and nitrate (Döscher et al., 1995; Schwikowski et al., 1999a), ammonium (Döscher et al., 1996), carbonaceous aerosols (Lavanchy et al., 1999; Thevenon et al., 2009), trace elements such as lead, copper, cadmium, zinc, plutonium (Barbante et al., 2004; Gabrieli and Barbante, 2014; Gabrieli et al., 2011; Schwikowski et al., 2004) and organic pollutants (Gabrieli et al., 2010; Kirchgeorg et al., 2013). Temporal variability of mineral dust including long-range transported dust from Africa was investigated by Bohleber et al., (2018), Gabbi et al., (2015), Gabrieli and Barbante, (2014), Wagenbach and Geis, (1989) and Wagenbach et al., (1996) using calcium, sulphate, iron and barium as mineral dust proxies.

## 2.2 Analytical methods

The top 57.2 m of CG03B comprising 1635 discrete samples (cross section area 1.9 x 1.9 cm) were analyzed at the Paul Scherrer Institut (PSI) between May and July 2015 (Table 1) achieving sub-annual resolution since the preindustrial (i.e. 1741 AD). Concentrations of rBC were determined with a Single Particle Soot Photometer (*SP2*, Droplet Measurement Technologies, (Schwarz et al., 2006)) and a jet (*APEX-Q,* Elemental Scientific Inc.) nebulizer to aerosolize the aqueous samples (Wendl et al., 2014). BC analysis was done at approximately 2 cm water equivalent depth resolution. Freshly cut samples stored in polypropylene vials were melted at room temperatures, sonicated for 25 minutes and measured immediately using an auto-sampler (*CETAC ASX-520*). Liquid sample flow rates typically varying within ±10% were measured routinely to ensure constant aerosolization efficiencies. Capillaries delivering samples to the nebulizer were rinsed

at least on a daily basis with 3% nitric acid for 10 minutes. For external calibration we used rBC standard solutions (*Aquadag®*, Acheson Inc., Wendl et al. 2014 for details) freshly prepared and directly analyzed at concentrations from 0.1 to 50 ng g$^{-1}$. To demonstrate the reproducibility of the rBC measurements and the robustness of the method we performed 387 replication analyses using parallel ice-core sections that comprised 20% of the length of the original analyses. The upper

11.9 m of CG15, in total 276 samples, were similarly analyzed. Following rBC analyses, we determined the concentrations of major ions (Na$^+$, NH$_4^+$, K$^+$, Ca$^{2+}$, Mg$^{2+}$, Cl$^-$, NO$_3^-$, SO$_4^{2-}$) using ion-chromatography (*850 Professional IC,* Metrohm).

For the parallel ice core CG03A a wide range of additional elemental and chemical components of aerosols had been analyzed using ion-chromatography (IC, *Dionex*) and inductively coupled plasma mass spectrometry (ICPMS, *Agilent 7500*), enabling ice-core dating (Jenk et al., 2009) and detailed characterization of dust and pollution aerosols (Gabrieli and Barbante, 2014). Trace element analyses were performed at University of Venice using the ICPMS in continuous flow mode

achieving an effective sampling resolution of approximately 0.5 cm water equivalent (Gabrieli and Barbante, 2014). Here we present measurements of trace metals (i.e., bismuth, Bi), typically emitted by coal burning and other industrial processes (McConnell and Edwards, 2008) and of chemical tracers (i.e. calcium, Ca$^{2+}$) typically enriched in mineral dust originating predominantly from the Saharan deserts and constituting a second potential source for light-adsorbing impurities present in

Alpine glacier ice.

## 2.3 Ice core dating

The CG03B ice core was dated against the chronology of the CG03A core (Jenk et al., 2009) using the major ion records obtained for both cores to align the records. In total 221 stratigraphic links were established between these two records between 1741 and 2003 AD which is close to the number of annual layers identified originally in CG03A. Linear

interpolation was used to date the ice between the stratigraphic tie-points. Differences in the depths for common time markers are found to be less than 13 cm at most (Supplementary Table S1). The chronology of CG03A (Supplementary Fig. S1; Table S1) was originally derived by annual-layer-counting predominantly using the NH$_4^+$ record and constrained by absolute age markers from volcanic eruptions, nuclear weapon testing, historic Saharan dust events and $^{14}$C dating of insoluble organic carbon in the deeper core sections (Jenk et al., 2009; Sigl et al., 2009; Uglietti et al., 2016). Previously

identified volcanic horizons (i.e., Katmai 1912, Tambora 1815, Laki 1783) were corroborated in the new CG03B records using SO$_4^{2-}$ concentrations together with the ratio of SO$_4^{2-}$/Ca$^{2+}$. Additional volcanic signatures (e.g., in 1809) potentially relating to a large eruption of unknown origin in 1809 AD were detected in CG03B but have not been used to further constrain the timescale. Constrained by historic events during the beginning and end of the 19$^{th}$ century, maximum age uncertainties are conservatively estimated to be ±5 years at most during the mid-19$^{th}$ century (Jenk et al., 2009).

## 2.4 BC emission inventories

In the absence of direct BC measurements during the past hundreds of years, gridded emission inventories from bottom-up approaches, (e.g., Bond et al., (2007); Lamarque et al., (2010)), are widely used to estimate emissions and aerosol loading.

These are the final products of a wide range of estimates of activity (e.g., fuel consumption) combined with emission factors (e.g., grams of BC emitted per mass of fuel burned derived from controlled burning of fuel types under laboratory conditions) and therefore carry large uncertainties (Bond et al., 2013). While the general emission trends from these inventories could be confirmed through comparison to existing ice core reconstructions (Jenk et al., 2006; Junker and Liousse, 2008; Lavanchy et al., 1999), a more detailed evaluation was hampered by the relatively large error ranges inherent in both these reconstruction approaches. For this study, we deploy the BC emission estimates from fossil-fuel and biofuel burning (available at 5 year resolution) from Bond et al. (2007) for 1) the OECD countries in Europe and 2) the mean of the grid cells 45-47°N and 6-9°E (available at 10 year resolution) encompassing both our ice-core study site and the locations of various glacier length reconstructions in the Western Alps (see section 2.5; Fig. 1).

## 2.5 High-resolution glacier length histories from Western Alps

Glacier fluctuations in the European Alps are among the best documented worldwide, since glaciers are situated in densely populated areas. Painter et al., (2013) used five glaciers from the Western and Eastern Alps to analyze the 19[th] century changes of their terminus positions, with Unterer Grindelwald glacier providing the densest observation frequency during the 19[th] century among these glaciers. For this study, we compile four glacier length reconstructions from the Western Alps (all situated in close proximity to the ice-core site) including Mer de Glace, Oberer Grindelwald and Unterer Grindelwald glacier, and Bossons glacier (Nussbaumer and Zumbühl, 2012), the latter offering the highest observation density during the mid-to-late 19[th] century with annual data coverage between 1850 and 1899 AD of 78%. To analyze trends in glacier length variability relative to the increase of industrial black carbon deposition at Colle Gnifetti and Fiescherhorn (Jenk et al., 2006), we filled missing terminus position data by linear interpolation and constructed a stacked glacier length curve by averaging the terminus positions of all four glaciers.

## 2.6 Time-of-emergence (ToE) analyses

To determine the timing when rBC concentrations exceeded their natural variability, suggesting an additional, industrial emission source, we performed a time-of-emergence (ToE) analysis on annually averaged rBC concentrations. The ToE is formally defined by Hawkins & Sutton (2012) as the mean time at which the signal of change emerges from the noise of natural variability. We followed the methods of Abram et al., (2016) in defining the threshold of emergence value as the earliest occurrence where the signal-to-noise ratio exceeds the value 2 (industrial rBC signal is distinguishable from zero at a 95% confidence level). We consider the time period 1741 to 1840 AD as preindustrial reference period during which human emissions of light absorbing rBC in Central Europe was minimal, restricted to occasional forest fires and residential wood burning (henceforth summarized as biomass burning BB). To discriminate large rBC values caused by BB within the preindustrial period (1741-1850 AD) we employed a fire detection algorithm adapted from Fischer et al., (2015) and replaced the correspondent BC concentration values for detected "fire activity" years with their 11-year running medians ($BC_{no\ BB}$). The ToE assessment was carried out varying the length of the pre-industrial reference period and the degree of

smoothing (running mean) applied to the record to determine the underlying signal. Reference periods from 15 to 100 years in length were used beginning in 1741 (i.e. shortest reference period is 1741-1765 AD and longest reference period is 1741-1840 AD) were used to determine the mean and $+2\sigma$ level of natural variability. A running mean of the same length as used for the reference period was applied to determine the signal and ToE was assigned to the year when the signal first

permanently exceeded the $+2\sigma$ value of the reference period. This method results in a distribution of ToE estimates that reflect uncertainty based on methodological choices in defining the signal and noise values used to define ToE. We assume the uncertainty in ToE to be independent from the ice-core dating uncertainty, and use their root sum square as the total uncertainty estimate of the emergence of enhanced industrial BC emissions. ToE analysis was similarly applied to total rBC (without discriminating BB years) and to $Ca^{2+}$ as a proxy for Saharan dust deposition, the latter having also high abundances

of light-absorbing minerals such as hematite $Fe_2O_3$ (Linke et al., 2006). To evaluate the sensitivity of the choice of detection method for ToE we also employed a Bayesian Change Point algorithm (Ruggieri, 2013) based on a linear regression model to determine the timing for a significant change point (equivalent to the emergence) within the rBC time series.

## 3. Results

### 3.1 BC and source tracer variability since 1741 AD

The synchronized records of CG03, CG08 and CG15 (Supplementary Fig. S2) provide a continuous record of long-term changes of rBC deposition at this site from the preindustrial (1741 AD) into the most recent past (2015 AD, Fig. 2). Deposition histories for major aerosol species are highly reproducible in the two parallel CG03 ice cores (Supplementary Figs. S3, S4), indicating minimal adverse effects of snow drift and spatial variability of aerosol deposition present at these spatial scales. Measured rBC values at CG03B vary strongly on intra-annual timescales, with this variability being

superimposed on longer-term trends. Replicate analyses performed at the end of the measurement campaign confirm that the original measurements performed over 2 months are highly reproducible over a concentration range of almost three orders of magnitude (Fig. 2, Supplementary Fig. S5). Between 1975 and 2015 (N>12 samples per year) the highest values of >15 ng g$^{-1}$ (90%-tile) are typically recorded during the summer months reflecting increased deposition of aerosol species on the glacier during times when the planetary boundary layer reaches higher than the drilling site at 4450 m (Lugauer et al., 1998).

During the remaining season of snow accumulation, the ice-core site is situated within the free troposphere with rBC concentrations significantly lower with approximately 0.9 ng g$^{-1}$ (10%-tile). Due to common transport and deposition, co-variability of rBC with other species at intra-annual resolution is indicated by significant *Pearson's* correlation coefficients (p<0.001, one sided, N=696, $Na^+$, R=0.35; $NH_4^+$, R=0.63) for the period of the most recent 40 years).

To analyze long-term rBC variability we calculate annual mean values by averaging all rBC values within the respective

calendar year (Fig. 3). Excluding occasional rBC spikes (>4 ng g$^{-1}$) the mean rBC concentration in the CG03 core during the preindustrial (i.e., 1741-1850 AD) was 2 ng g$^{-1}$, followed by a small (2-fold) increase to approximately 4 ng g$^{-1}$ during the last two decades of the 19$^{th}$ century (1880-1899 AD). Maximum rBC concentrations of 10 ng g$^{-1}$ exceeding five times the

preindustrial values are recorded between 1910-1920 and again 1933-1945, with a short decline between 1921-1932 (6 ng g$^{-1}$), plausibly explained by a drop in industrial rBC emissions following the economic crisis between the two world wars. Since 1950, rBC concentrations remained elevated and only started to drop significantly after 2000 AD. Since then, concentrations vary around 5 ng g$^{-1}$, still more than twice as high as during the preindustrial period. We summarize median rBC concentrations in Table 2 together with other relevant ice-core source tracers and their main emission sources for the preindustrial and for two periods labeled after their main fossil-fuel source as coal "*COAL*" (1901-1950) and petroleum products "*PETROLEUM*" (1951-1993), respectively. 1993 AD marks the end of the time period for which continuous trace element analyses were performed.

In the preindustrial period, ammonium (R=0.75, p<0.0001) and nitrate (not shown) are strongly correlated with rBC indicating that BC was associated to (natural or anthropogenic) biomass burning emissions (Fig. 4; Supplementary Fig. S6). Sulphate (SO$_4^{2-}$) concentrations started to rise in 1900 AD, with this timing well constrained by the Saharan dust event of 1901 AD (Oeschger, 1977; Wagenbach and Geis, 1989)). Throughout the *COAL* era SO$_4^{2-}$ concentrations are strongly correlated with rBC, with this correlation becoming weaker during the *PETROLEUM* era. Heavy metals such as lead (Pb) and Bi show high relative enrichments comparable to rBC in particular from 1910 to 1950 AD indicating an increased association of BC with anthropogenic fossil-fuel emissions (e.g., coal burning) during that time (Table 2). None of the four discussed industrial pollution tracers (rBC, SO$_4^{2-}$, Pb, Bi) shows pronounced increases in concentrations starting in the mid-19[th] century. In agreement with other dust records from Colle Gnifetti (Bohleber et al., 2018; Wagenbach and Geis, 1989), we observe no enhanced mean (or frequency) of mineral dust deposition throughout the 19[th] century (Supplementary Figs. S1, S4). Only during the past three decades (1975-2015 AD), does the dust activity appear to be anomalously high with respect to the long-term variability (Supplementary Fig. S6) which is thought to relate to increased drought conditions in the main dust source regions in Northern Africa (Moulin and Chiapello, 2006).

## 3.2 Comparison to other ice-core BC records

Few other ice-core records contain precisely dated information about Central European industrial BC emissions for the 19[th] century. Previous determinations of EC concentrations from the Colle Gnifetti ice core with various methods are characterized by coarse resolution and unknown reproducibility (Lavanchy et al., 1999; Thevenon et al., 2009). The Fiescherhorn FH02 ice core obtained 70 km north of CG03 in the Bernese Alps (3900 m a.s.l., 46°33'03"N; 08°04'00"E) provides total EC concentrations from 1650-1940 AD at 5-10 year resolution (Fig. 3b) (Jenk et al., 2006) and analyses were recently completed at annual resolution until 2002 AD (Cao et al., 2013; Gabbi et al., 2015). The low-resolution record from Fiescherhorn (Jenk et al., 2006) and Colle Gnifetti (Thevenon et al., 2009) were used by Painter et al., (2013) as input for their model study on potential changes of 19[th] century snow-albedo (note that in Fig. 2 in Painter et al. (2013), the labelling of the two cores was swapped). Despite the coarse resolution, the overall EC trend from Fiescherhorn ice core is closely reproduced by the new rBC record from CG03 (R=0.71, p<0.005) (Fig. 3 c,d). Differences in absolute concentrations by approximately a factor of four can be understood to reflect a difference in elevation (FH02 is situated 500 m lower in

elevation and closer to the emission sources) and a difference in the analytical methods employed, with thermo-optical methods resulting in consistently higher values compared to photometric determination of rBC (Currie et al., 2002; Lim et al., 2014) (also see Sect. 4.1). Nevertheless, the common three-step increase of concentrations occurring around 1875 AD, 1900 AD and 1940 AD, respectively, strongly supports the interpretation that both ice cores capture a common signal of industrial BC emissions increase driven by technological and economic developments in Central Europe.

In contrast, the low resolution EC record of Thevenon et al., (2009) shows a markedly different variation in time. It was produced from non-equidistant samples, was obtained with a not-validated method, and used a preliminary version of a modelled Colle Gnifetti ice-core chronology later published by Jenk et al. (2009). Since the age-model used by Thevenon et al (2009) was not forced to intersect with the absolute dated reference horizons, it is biased from the annual-layer dated chronology by on average 14 years, (7-18 years) during the 19$^{th}$ century. The final chronology dates the early ~1850 AD increase in EC in the 1830s, which would imply an early start of industrialization. This is neither consistent with the new rBC record from Colle Gnifetti or with the Fiescherhorn EC record nor with rBC records from Greenland (see below). Given that mean 16$^{th}$ century EC concentrations reach comparable levels than during the peak industrial era of the 20$^{th}$ century, we assume that the Thevenon et al., (2009) EC record suffered from methodological biases, probably related to the presence of mineral dust (i.e., high EC concentrations occurred often in samples with high dust concentrations, see Fig. 3 in Thevenon et al., (2009)).

The non-BB rBC record (CG03 rBC$_{no\ BB}$) was compared to an equivalent stacked rBC$_{no\ BB}$ record obtained from four ice cores (Summit 2010, D4, NEEM-2011-S1, TUNU2013; Fig. 5) from Greenland (Keegan et al., 2014; McConnell et al., 2007; Mernild et al., 2015; Sigl et al., 2013; Sigl et al., 2015) acknowledging that these Greenland ice cores capture a mixture of emissions from both Northern America and Europe (Bauer et al., 2013; Hirdman et al., 2010; Lamarque et al., 2013; McConnell et al., 2018). With high snow accumulation rates and analyzed at high-resolution, absolute dating uncertainties for these records are estimated to be better than ±1 year which provides us with another independent, high-precision age constraint for the onset of increased industrial BC emissions from Europe.

Whereas absolute concentrations are, depending on the specific industrial pollutant (BC, Bi), a factor of 2 to 4 lower in Greenland, the long-term trends are remarkably similar between the Greenland stack and CG03 (Fig. 4d, Fig 5). Overall, concentrations of major industrial pollutants appeared to have increased earlier by roughly 10 years (in 1890 AD) in Greenland compared to the Alps, most prominently visible in bismuth. This delay is consistent with industrialization having accelerated earlier in North America (McConnell and Edwards, 2008) than in the major Central European countries (e.g., Germany, Italy, France). The maximum in industrial BC emissions were synchronous between Greenland and the Alps, both peaking at approximately 1915 AD. Differences exist in the long-term trends of rBC since the early 20$^{th}$ century maximum, with Greenland values closely approaching preindustrial levels while remaining elevated in the Alps.

### 3.3 Comparison with BC emission inventories

We compare our ice-core based deposition history with estimated emissions of BC from fossil-fuel and bio-fuel burning (Bond et al., 2007) which form the main input for simulating BC climate effects (direct, indirect aerosol, and ice-albedo forcing) on past climate (Flanner et al., 2007; Lamarque et al., 2013; Shindell et al., 2013). We notice that the general structure of BC and EC from the two Alpine ice cores closely resembles estimated BC emissions for both OECD Europe and the Western Alpine region taken from the bottom-up inventory of Bond et al., (2007) (Fig. 6). Interpreting the ice-core long-term trends as proxies for atmospheric burden (or emissions, respectively) we identify three major differences between these datasets. First, the increase in BC to its early 20$^{th}$ century maximum as deduced from the ice cores occurred in two subsequent steps, whereas the emission inventory implies a more gradual increase throughout the 19$^{th}$ and early 20$^{th}$ century. Second, the BC inventory emissions remain fairly steady at high levels from 1910 to 1950 AD with no decrease between the two world wars as in both Alpine ice cores, and to some degree also in the Greenland ice cores (Fig. 5). Third, the emission inventories suggest that BC emissions dropped significantly since the 1960s and reached for OECD Europe preindustrial levels by 1980, whereas BC ice-core concentrations remained clearly above their preindustrial values until the very recent past. Median concentrations from 1980 onwards are still 3-fold at CG03 and 1.5-fold at FH02 compared to pre-industrial 1741-1850 AD levels, respectively. Table 3 summarizes BC emission estimates based on inventories and mean ice-core rBC concentrations centered at 1850, 1915, 1975 and 2000 AD for CG03, FH02 and the a Greenland ice-core stack, respectively.

### 3.4 The timing of industrial BC deposition and glacier lengths variations in Europe in the 19$^{th}$ century

To test the plausibility of an ice-albedo effect to force (or at least contribute) to the glacier length reductions occurring during the 19$^{th}$ century we here examine the exact timing of industrial BC deposition at Colle Gnifetti and Fiescherhorn. For the latter, we interpret the sharp increase in EC (Fig. 3) in the sample dated to the years 1875 to 1879 AD (Jenk et al., 2006). Assuming conservatively that the increase occurred at the start age of this discrete sample and considering a dating uncertainty of ±5 years provides us with a lower bound for the earliest occurrence of enhanced BC deposition of 1870 AD. ToE analysis for the CG03 rBC$_{no\ BB}$ record identifies the year 1875 AD [5-95% range: 1870-1882 AD] as the time of emergence of industrial BC deposition with a conservative dating uncertainty of ±5 years (Fig. 7, Table 4). Since ToE and dating uncertainty are independent, we estimate the absolute uncertainty range in the timing of industrial BC deposition at CG03 as 1868-1884 AD (5-95% range). The Bayesian Change Point algorithm returns virtually the same result with the highest change point probability in 1876 AD (Table 4). The median timing of industrial BC deposition at the four Greenland ice-core sites is 1872 AD (ToE analysis) or 1891 AD (Bayesian Change Point), respectively, in good agreement with the Alpine ice cores. Using the precisely dated D4 ice core and vanillic acid to discriminate forest fire emissions, McConnell et al. (2007) gave 1888 AD as their best estimate for industrial BC emergence, closely matching our best estimates employing ToE analysis (1878 AD) and the Bayesian changepoint method (1891 AD), respectively (Supplementary Fig. S7)

The four high-resolution glacier length records indicate that in 1875 AD, these glaciers had already completed the majority of their total cumulative length reductions (i.e. maximum to minimum front position) of the second half of the 19th century. Bossons had experienced 100%, Oberer and Unterer Grindelwald 83% and 74%, respectively, and Mer de Glace 79%, of their cumulative length losses (Table 4; Fig. 6e), with differences likely explained by the different size and topography of the individual glaciers (Lüthi, 2014). Consequently, the stacked record of all four glacier terminus position curves reveals that the highest annual mean glacier length reduction rates of >40 m yr$^{-1}$ occurred during the 1860s, when BC concentrations in both ice-cores were still indistinguishable from their natural background levels (Fig. 7). During time-of-emergence of industrial BC deposition in 1875 AD the stacked glacier record had experienced 83% [52-92%] of its entire cumulative glacier retreat from the maximum 1850s terminus positions. ToE analyses performed for total BC (ToE: 1890 AD) and for calcium (ToE: 1986 AD) are equally inconsistent with a mid-19th century emergence of the light-absorbing impurity content on Alpine glaciers outside the range of natural variability (Supplementary Figs. S8, S9).

## 4. Discussion

### 4.1 Alpine glacier fluctuations, industrial BC and post-volcanic cooling

For the first time, we are able to examine a continuous, well-dated record of BC from the preindustrial into the most recent present at sub-annual resolution (Fig. 2). Highly reproducible rBC measurements mirror the low-resolution EC record obtained from the nearby Fiescherhorn ice core (Fig. 3). We interpret this as evidence that these ice cores detect a common signal of the atmospheric BC burden since the preindustrial from anthropogenic emissions of BC by industrial and transport related activities. Other source tracers co-analysed with BC allow attribution of changes of the main emission sources to the observed trends (Fig. 4). During the preindustrial (1741-1850 AD) CG03 rBC concentrations were low with episodic spikes co-registered with ammonium attributed to anthropogenic or natural biomass burning sources. Only later in the 19th century did concentrations of rBC and other industrial pollutants (e.g., Bi, Pb, $SO_4^{2-}$) typically emitted by coal burning start to increase significantly, with 1875 AD identified as the best estimate and 1868 AD as a very conservative lower bound for the timing of the earliest emergence of these emissions from background variability. Greenland ice cores, which also capture emissions from Europe, are consistent with our finding that no major increase of BC and other typically co-emitted industrial tracers (e.g. Bi) occurred before 1870 AD (Fig. 5). Mineral dust deposition at CG03 does not show relevant long-term trends during the 19th century.

The new combined evidence strongly contradicts the previous key assumption of a synchroneity between glacier retreat in the European Alps and BC increase in the 19th century in apparent support of the hypothesis that industrial BC emissions have forced accelerated glacier melt through a snow-albedo feedback (Painter et al., 2013). With 82% [52%] of the glacier length reductions having already taken place at the best [earliest] estimated time of emergence of industrial BC deposition, these are unlikely to have been forced by anthropogenic soot emissions (Fig. 7). The discrepancy in the temporal relation between our results and those of Painter et al., (2013) are in part explained by the low resolution in their deployed glacier

length records in the 19[th] century (i.e., Rhône, Argentière), that tend to smooth the actual terminus position curves between 1850 and 1900 AD but also by the limited quality of the BC records available at that time (see Sections 1. and 3.2). As shown in Fig. 8, retreat rates of the terminus positions from high-resolution glacier observations were much stronger between 1850 and 1875 AD, than they were between 1875 and 1900 AD. Moreover, when industrial BC emissions reached their overall maximum values in the 1910-20s, indicated by ice-core BC concentrations exceeding five-times their preindustrial values, Alpine glaciers showed no indications of further retreat, but were instead advancing again (Fig. 7, Fig. 8). Since our rBC measurement technique is less sensitive to "brown carbon" and mixed-component aerosols and larger compounds outside the detectable size range (up to mass-equivalent diameter of 800 nm) such as produced by burning low quality coal or inefficient coal combustion (Sun et al., 2017) this record alone cannot rule out a potential role for other light-absorbing aerosols. However, these fractions are entirely accounted for by the method applied for the EC record from FH02 showing very good agreement with the rBC record from Colle Gnifetti (Section 3.2) and remaining at pre-industrial values until 1875 AD. Thus, other factors than changes in surface snow albedo appear to have dominated mass balance and glacier length variability of these European glaciers until at least 1875 AD, of which temperature and seasonal precipitation distribution are widely considered the most important (Steiner et al., 2008; Zumbühl et al., 2008). The previously made claim (Painter et al., 2013; Vincent et al., 2005) that precipitation and temperature variability alone were insufficient to explain the observed glacier length variability, has been recently challenged by Lüthi (2014) and Solomina et al., (2016), who demonstrated that on a regional to global scale summer temperatures are the most important parameters determining glacier mass balance variability. Previously underestimated due to an "early-instrumental warm bias" (Böhm et al., 2010; Frank et al., 2007) new surface air temperature (SAT) reconstructions based on early instrumental records, historical documentary proxy evidence and tree-rings now all give evidence that the early 19[th] century was exceptionally cold in Central Europe in a long-term context (Brohan et al., 2012; Büntgen et al., 2011; Luterbacher et al., 2016) (Fig. 9, Supplementary Fig. S10). A strong negative radiative forcing resulting from at least five large tropical eruptions between 1809 and 1835 (Sigl et al., 2015; Toohey and Sigl, 2017), in tandem with the Dalton solar minimum (Jungclaus et al., 2017; Usoskin, 2013) appeared to have forced the glaciers to strongly advance until the 1850s, in some cases probably far outside their range of typical long-term natural variability (Fig. 8). Similarly, later glacier advances (e.g. in 1890s and 1910s) followed other major volcanic eruptions including Krakatao (1883) and Katmai (1912).

A strong role of volcanic forcing is supported by a consistent strong coherence of glacier expansions following clusters of volcanic eruptions throughout the past 2,000 years (Le Roy et al., 2015; Solomina et al., 2016). The stratospheric aerosol burden for the time window 1600-1840 AD was 40% larger than during the entire Common Era (Sigl & Toohey, 2017) with volcanic eruptions frequently forcing cold spells and glacier advances (e.g., in 1600s, 1640s, 1820s, 1840s) in the Alps (Fig. 9) and elsewhere (Solomina et al., 2016). Increased summer precipitation during cool post-volcanic summers may have additionally contributed to a more positive mass balance (Raible et al., 2016; Wegmann et al., 2014) plausibly enforced by a positive albedo feedback loop resulting from increased snow cover in the Alps. The glaciers' initial and more or less synchronous retreat from the maximum terminus positions starting at 1860 can thus be understood as a delayed rebound back

to their positions they had before the radiative perturbed time period 1600-1840 AD (Fig. 9), and an additional decrease of snow albedo from the deposition of BC is considered not to be needed to explain these observations (Lüthi, 2014). The specific extent to which early anthropogenic warming (Abram et al., 2016), changes in atmospheric modes (Swingedouw et al., 2017) including the Atlantic Multidecadal Oscillation (AMO) (Huss et al., 2010), or to some extent snow-albedo feedbacks from increasing light-absorbing aerosol deposition towards the end of the 19[th] century may have contributed to the overall glacier length variability in the European Alps throughout the 19[th] century remains difficult to determine. To confidently attribute and quantify the contribution of natural and anthropogenic forcing to observed glacier changes will require to reconcile early instrumental and proxy climate data (Böhm et al., 2010; Frank et al., 2007), and to use ensembles of simulations with a hierarchy of model complexities to decompose the relative contribution of volcanic eruptions, light-absorbing impurities such as BC or other compounds (e.g., brown carbon, mineral dust) and other potential natural or anthropogenic contributions (Goosse et al., 2018; Zekollari et al., 2014).However, this is out of the scope of this paper focusing on the timing of events, the importance of dating accuracy and analytical repeatability of measurements in the presented ice core records. By demonstrating a time-lag between industrial BC increases inferred from two ice cores and observed glacier terminus positions from four European glaciers, we cast doubt on an initiating role of industrial BC for the mid-19[th] century glacier retreat in the Alps as previously suggested in Painter et al. (2013). Its influence on glacier albedo and subsequent effect on glacier retreat has to be considered carefully for the 20[th] century when BC snow concentrations became significantly higher.

## 4.2 Constraints on Central European BC emissions

While the CG03 rBC time series reproduces well the general major emission trends from gridded BC emission inventories (Fig. 6) it provides additional structure that is currently not captured by the BC inventories. This includes a stepwise increase of BC rather than linearly rising emissions; a short reduction of emissions between the two World Wars likely related to global economic depression (Gabrieli and Barbante, 2014; Schwikowski et al., 2004) and smaller reductions since the 1960s as opposed to the BC emission inventories. Similar to Europe, Greenland ice-core BC records also do not support the idea of a gradual increase in BC since 1850 AD but show a very rapid increase of emissions around 1890 AD, suggesting that the emission inventories data before 1900 AD may be biased. This is plausible given the small number and incomplete nature of consumption and technology-related records contributing to these inventories during this time (Bond et al., 2007; Bond et al., 2004). Since many datasets (e.g., refinery outputs) during the 19[th] century were only available for the USA and with extrapolation backward in time applied often when specific data was unavailable (Bond et al., 2007) it is not surprising that the BC emission trend in Europe (and other regions) more or less closely follows that for North America prior to 1900 AD (Bond et al., 2007; Lee et al., 2013a). The discrepancy between the inventory-estimated and the much lower ice-core indicated reduction of BC in CG03 since 1960s is striking. This mismatch suggests that the measures taken to reduce the release of BC into the atmosphere may not have been as efficient as the energy-consumption data suggests. This may hint that the emission factors (e.g., BC emitted per fossil-fuel-unit burned) are frequently reported as too low in these inventories,

which in the light of the Volkswagen emissions scandal revealed in 2015, seems at least a plausible scenario. A comparable offset had been recently also noted between modelled emissions and CG03 nitrate and ammonium records between 1995 and 2015 (Engardt et al., 2017) suggesting that besides rBC also $NO_x$ and $NH_3$ emissions may not be adequately accounted for in present-day emission inventories. Greenland ice cores capturing mostly emissions from North America (where diesel engines play a minor role compared to Europe) in contrast to the alpine ice cores show a very pronounced decrease of BC during the second half of the 20th century to almost preindustrial values at present (Keegan et al., 2014; McConnell et al., 2007). While providing more realistic estimates of carbonaceous particle emissions from gasoline and diesel engines remains an area of ongoing research (Gentner et al., 2012; Gentner et al., 2017; Platt et al., 2017) more records are certainly required from other suitable sites in Europe to elude the sources of this late 20th century mismatch. Further reducing uncertainties in ice-core BC records is of eminent importance in assessing the accuracy of emission inventories and of particular interest since all state-of-the-art coupled aerosol climate models use gridded BC emission inventories as input parameters for their simulations (Lamarque et al., 2010; Lee et al., 2013a; Shindell et al., 2013).

## 5. Conclusion

Industrial black carbon believed to be emitted in large quantities starting in the mid-19th century had been suggested as the key external forcing responsible for an accelerated melting of European glaciers through enhancing ice-albedo and subsequent ablation (Painter et al., 2013). We examined this interpretation by presenting new, highly resolved, well replicated ice-core measurements of refractory black carbon, mineral dust, and distinctive industrial pollution tracers from the Colle Gnifetti ice core in the Alps covering the past 270 years. The comprehensive suite of elemental and chemical species co-analyzed enabled to elucidate characteristic source profiles to disentangle industrial from biomass burning sources for BC. The precisely dated ice core allowed precise comparison of the timing of observed acceleration of glacier retreat in the mid-19th century with that of increased deposition of black carbon on the glaciers caused by the industrialization in Europe. Reproducing closely the main structure of the Fiescherhorn EC record (Jenk et al., 2006), our study suggests that at the time when European rBC emission rates started to significantly increase (only after 1870) the majority of Alpine glaciers had already experienced more than 80% of their total 19th century length reduction. Therefore, we argue that industrial BC emissions and subsequent deposition on Alpine glaciers are unlikely to be responsible for the rapid initial deglaciation at the end of the Little Ice Age in the Alps. Much more plausible appears an alternative hypothesis in that glacier length changes throughout the past 2,000 years have been forced pre-dominantly by summer temperatures reductions induced by sulphuric acid aerosol forcing from large volcanic eruptions. In this sense, the retreat from the volcanically forced maximum glacier terminus positions starting in the 1860s can be seen as a lagged response of the cryosphere after the volcanic induced cooling had reached its maximum following a sequence of major tropical eruptions in 1809, 1815, 1823, 1831 and 1835 AD. Only after 1870 AD, when BC emissions started to strongly increase, snow-albedo impurity effects may have potentially contributed to the glacier length reductions. This new hypothesis is up for future testing now.

Much of the understanding of future climate change is based on computer simulations, but models used to predict future climate must be evaluated against past climate for accuracy (Hansen et al., 2007; Lamarque et al., 2013). Aerosols in climate models are mostly evaluated with observations from recent years to a few decades, time periods during which mitigation measures for air quality control were widely in place. Ice core records, possibly the only data sets to provide long-term historical information on aerosols, are therefore critical for model evaluations, especially during time periods of widespread air pollution in industrialized countries during the 19th century. Here we performed a first step towards this goal by developing a first continuous BC record from Central Europe covering the past 270 years that has the resolution, precision and reproducibility to serve in the future as a benchmark for climate models through dedicated model-data intercomparison (Koch et al., 2011; Lee et al., 2013b). With aerosol deposition at this site understood to depend on shorter timescales (e.g. inter-annual-to-decadal) also on atmospheric transport efficiency and the spatial distribution and conservation of snowfall, incorporating more BC records from this site into a stacked composite is expected to enhance the signal (i.e., burden or emissions) with respect to the noise caused by atmospheric transport, spatially varying snow accumulation and preservation. This should, therefore, be considered a main focus for future research together with developing comparable records from other suitable ice-core sites in the Alps.

**Data availability:** The CG03 ice-core data are available in the XXXX repository https://xxxxx.

**Acknowledgment:**

This work was supported by the Swiss National Science Foundation through the research program "Paleo fires from high-alpine ice cores" (CRSII2_154450/1). The authors thank Joe R. McConnell and Nathan Chellman for providing Greenland ice-core data; Matt Toohey for providing SAOD reconstructions; Samuel Nussbaumer for providing glacier length reconstructions; Fang Cao for providing the FH02 EC record; Carlo Barbante for partly funding the CG03 drilling campaign; Sabina Brütsch for ion chromatography analyses; all members of the CG ice core drilling expeditions in 2003, 2008 and 2015. We also would like to thank Thomas Painter and colleagues and the three anonymous reviewers for their comments and valuable input helping to improve the manuscript.

**Author contribution:**

Mi.S conceived this study, performed BC and ion analyses, developed age-models, analyzed data and wrote the paper; N.J.A performed ToE analyses; J.G. analyzed trace elements, D.O. helped processing ice cores and developed SP2 methodology, T.M.J. supervised the development of the SP2 method and led the 2015 CG ice-core drilling campaign, Ma. S. coordinated the project. Mi. S. led the manuscript writing with input from all coauthors.

**Competing interests:**

The authors declare that they have no conflict of interest.

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

**Tables and Figures:**

**Table 1:** Ice cores, parameters and analytical methods

| Ice Core | Time (AD) | Depth | Analyses | Instrumentation |
|---|---|---|---|---|
| CG03A | 1741-1993/2003 | 0-57m | Trace elements (Bi, Pb) / major ions | ICPMS, IC |
| CG03B | 1741-2003 | 0-57m | Major ions, rBC | IC, SP2 |
| CG08 | 1996-2008 | 0-10.5m | Major ions | IC |
| CG15 | 2001-2015 | 0-11.9m | Major ions, rBC | IC, SP2 |

**Table 2:** Median concentrations for selected chemical and elemental tracers from Colle Gnifetti during preindustrial (PI), during time periods dominated by the fossil-fuel sources coal (COAL) and petroleum products (PETROLEUM).

|  | 1741-1850 | 1901-1950 | 1951-1993 |  |  |  |  |
|---|---|---|---|---|---|---|---|
|  | PI | COAL | PETROLEUM | COAL/PI | PETROLEUM/PI | Main sources | Main emitters |
| BC | 1.9 | 7.2 | 6.6 | 3.7 | 3.4 | cb, ind, t, bb, ff | Natural + anthropogenic |
| $NH_4^+$ | 34 | 66 | 131 | 2.0 | 3.9 | a, bb, ff, b | Natural + anthropogenic |
| $SO_4^{2-}$ | 80 | 284 | 679 | 3.6 | 8.5 | ind, t, cb, d, vol | Natural + anthropogenic |
| Pb | 0.11 | 0.55 | 1.32 | 5.1 | 12.3 | t, ind, cb | Anthropogenic |
| Bi | 1.2 | 4.4 | 3.6 | 3.7 | 3.0 | cb, ind, t | Anthropogenic |
| $NO_3^-$ | 80 | 100 | 251 | 1.3 | 3.1 | ind, t | Natural + anthropogenic |
| $Na^+$ | 17.1 | 17.3 | 18.9 | 1.0 | 1.1 | ss, d | Natural |
| $Ca^{2+}$ | 68 | 102 | 118 | 1.5 | 1.7 | d | Natural |

All concentrations are median concentrations in ng g$^{-1}$, except Bi (pg g$^{-1}$); calcium, sulphate, sodium and nitrate values are the mean from both ice cores CG03A and CG03B. BC is from CG03B, all others are from CG03A. cb = coal burning; ind = industrial; t = traffic; bb = biomass burning; ff = forest fires; a = agriculture; b = biogenic; ss = sea salt; d = mineral dust; vol = volcano.

**Table 3:** Estimated BC emissions from Bond et al., (2007) for OECD Europe and Western Alps (45-47°N, 6-9°E) with median concentrations of CG03 rBC and FH02 EC during Preindustrial (PI, 1840-1860), during peak coal burning (1910-1950), peak petroleum burning (1970-2000) and Present Day (PD, 1995-2005); numbers in brackets indicate the increase relative to PI (in %) (see Fig. 6).

|  | Preindustrial (PI) 1840-1860 | "Peak Coal" 1905-1925 | "Peak Petroleum" 1960-1980 | Present Day (PD) 1995-2005 |
|---|---|---|---|---|
| **Ice cores [ng g$^{-1}$]** |  |  |  |  |
| CG03 | 2.3 | 8.4 (+270%) | 6.6 (+190%) | 6.6 (+190%) |
| FH02* | 9.9 | 34.8 (+250%) | 21.0 (+110%) | 14.1 (+43%) |
| Greenland Stack (N=4) | 1.4 | 5.2 (+270%) | 2.3 (+59%) | 1.8 (+24%) |
| **Emission inventory (*Bond et al., 2007*) [Gg yr$^{-1}$]** |  |  |  |  |
| OECD Europe | 292 | 793 (+170%) | 653 (+120%) | 352 (+20%) |
| Western Alps (45-47°N/6-9°E) | 3.2 | 8.9 (+180%) | 8.0 (+150%) | 5.7 (+79%) |

based on EC analysis

**Table 4:** Alpine glacier lengths during the emergence of industrial BC deposition (see Fig. 7)

|  | Time-of-emergence | total 5-95% range (±5 yrs dating uncertainty) | Bayesian change point | total 5-95% range (±5 yrs dating uncertainty) |
|---|---|---|---|---|
| Year | 1875 | [1868-1884] | 1876 | [1870-1881] |
| **% of total mid-19th century (i.e. 1850-1880 ) glacier length reduction completed** |  |  |  |  |
| Bossons | 100 | [78-100] | 100 | [83-100] |
| Mer De Glace | 79 | [37-89] | 84 | [56-92] |
| O. Grindelwald | 83 | [53-89] | 87 | [62-90] |
| U. Grindelwald | 74 | [54-100] | 76 | [63-94] |
| Stack (N=4) | 82 | [52-92] | 85 | [63-91] |
| Median (N=4) | 81 | [54-94] | 86 | [62-93] |

**Table 5:** Alpine glacier advances and volcanic eruption dates and resulting stratospheric aerosol properties (see Fig. 8)

| Major glacier advance phases (AD) | Cumulative glacier length change (m) | SAOD$_{30-90°N}$ rel. to 1900-2000 AD (%) | Major eruptions (SAOD$_{30-90°N}$>0.02) [Rank among all eruptions in 1800-2000 AD] |
|---|---|---|---|
| 1807-1820 | +450 | +320% | 1809 [2]; 1815 [1] |
| 1831-1854 | +126 | +58% | 1831 [3]; 1835 [9]; 1846 [14] |
| 1883-1893 | +115 | +109% | 1883 [5]; 1890 [10] |
| 1913-1924 | +154 | -20% +143% (lag -10 years) | 1902 [6]; 1907 [11]; 1912 [8] |
| 1967-1982 | +183 | 0% | 1963 [12]; 1975 [15]; 1982 [4] |

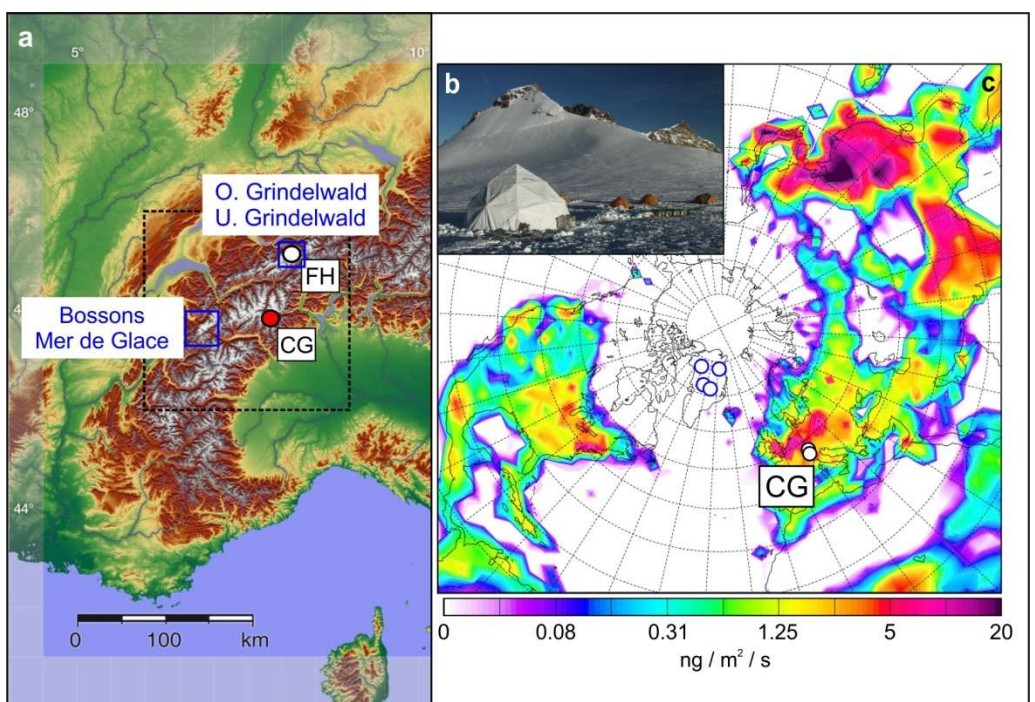

**Figure 1: a) Colle Gnifetti (CG) and Fiescherhorn (FH) ice-core drilling sites, high resolution glacier length reconstructions from the Bernese Alps (O. Grindelwald, U. Grindelwald) and French Alps (Bossons, Mer de Glace). Dashed rectangle envelopes the six 1° x 1° grids used for the comparison of gridded black carbon (BC) emission estimates (Bond et al., 2007) with ice-core BC concentrations (Fig. 6); Source: Perconte (Based on SRTM-Data) [CC BY-SA 2.5 (https://creativecommons.org/licenses/by-sa/2.5)], via Wikimedia Commons b) drilling site of the new CG15 ice-core at 4450 m asl (Source: M. Sigl); c) Alpine and Greenland ice-core drilling sites superimposed on present-day annual mean fossil fuel and biofuel BC emission estimates (adapted from Stohl (2006)).**

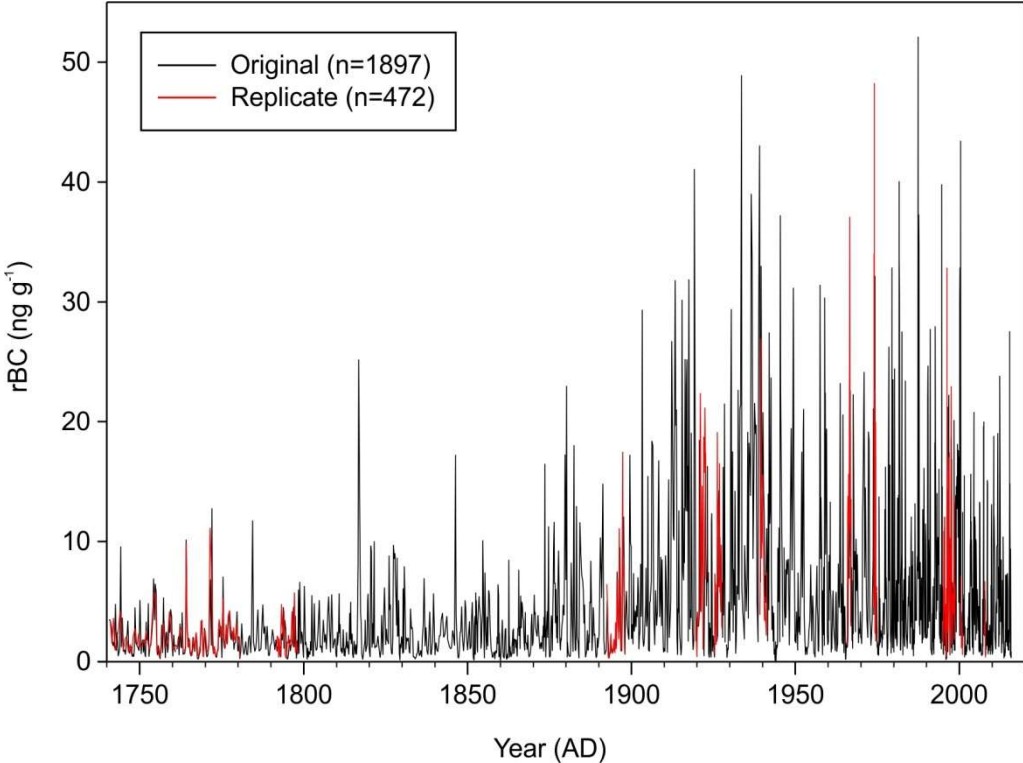

**Figure 2: Combined (CG03 and CG15) Colle Gnifetti rBC concentration record including the original analysis (black) and replicate samples from parallel core sections (red) between 1741-2015 AD.**

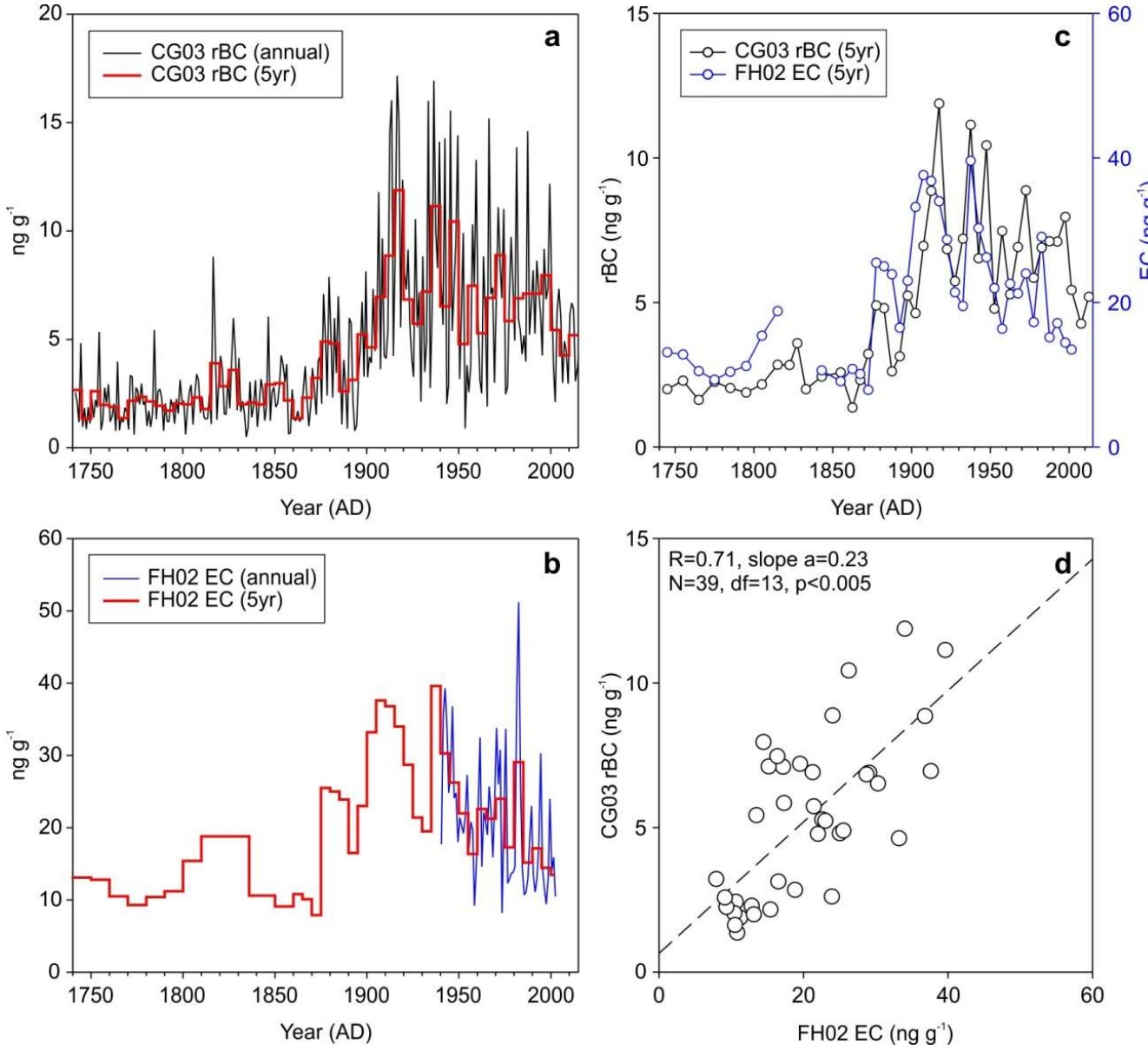

**Figure 3: a)** Colle Gnifetti CG03 rBC concentrations and **b)** Fiescherhorn FH02 elemental carbon (EC) concentrations (Jenk et al., 2006; Cao et al. 2013; Gabbi et al. 2015); **c)** comparison of CG03 and FH02 ice cores resampled to the FH02 sampling resolution of 5-10 years with **d)** linear fit and *Pearson's* correlation coefficient *R*=0.71, *P*<0.005 (adjusted for a reduced sample size owing to autocorrelation of the data sets) indicated.

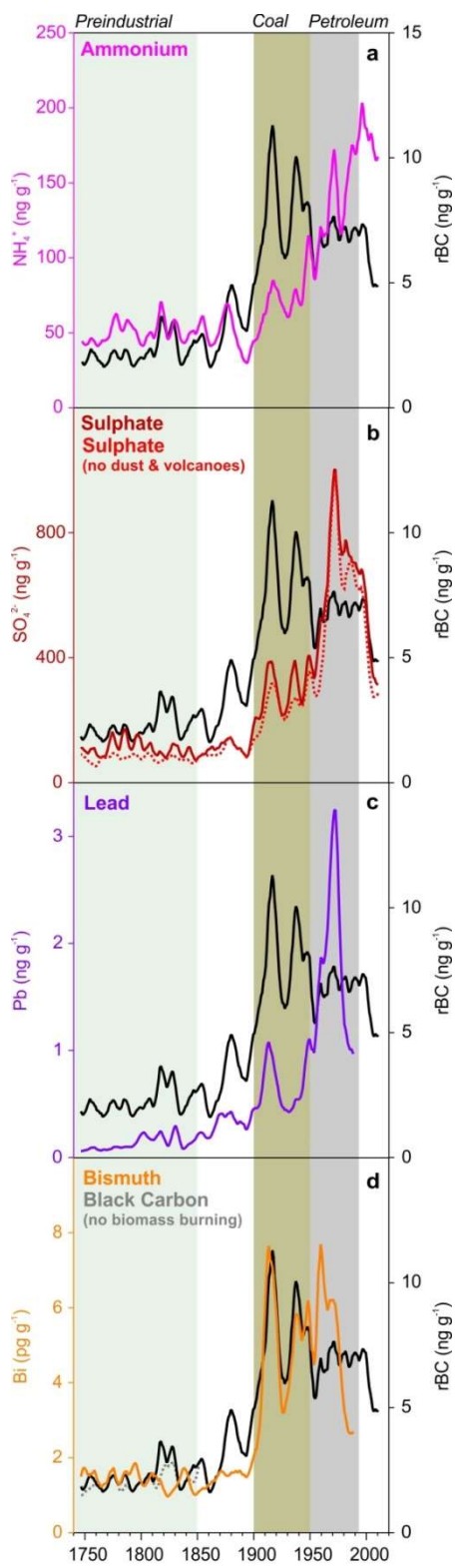

**Figure 4: Colle Gnifetti rBC record (black) compared with a) ammonium (NH$_4^+$), b) total sulphate (SO$_4^{2-}$) and sulphate corrected for contributions from volcanic emissions and Saharan dust (dotted line), c) lead (Pb) and d) bismuth (Bi) together with rBC corrected for biomass burning contributions (BC$_{no\ BB}$; grey dotted line). All records are 11-yr filtered. Marked are time periods dominated by preindustrial emissions (1741-1850 AD), coal burning (1901-1950 AD) and petroleum burning emissions (1951-1993 AD), respectively.**

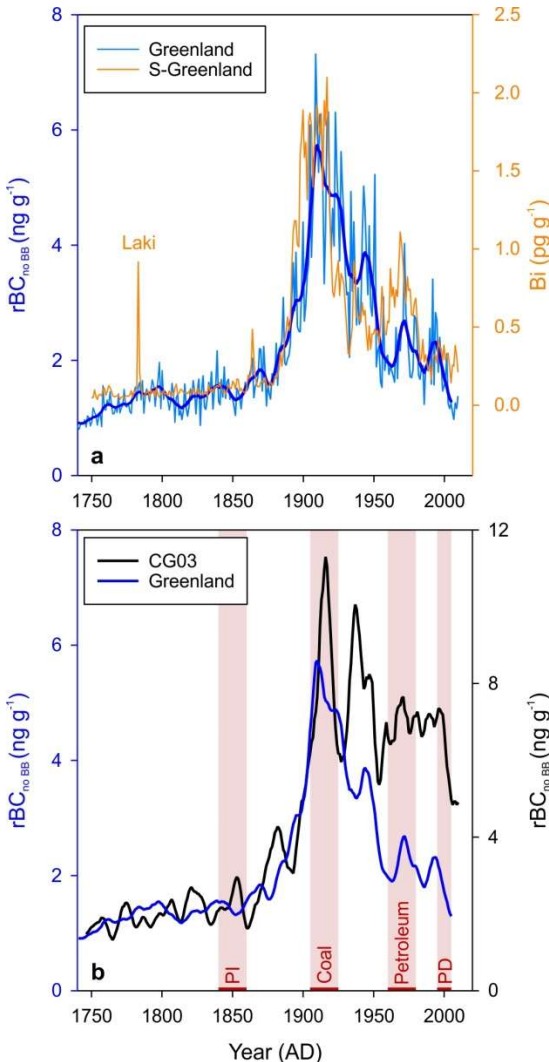

**Figure 5: a) Stacked rBC record from Greenland including the four ice cores NEEM-2011-S1, D4, TUNU2013 and Summit2010 (Keegan et al., 2014; McConnell et al., 2007; Mernild et al., 2015; Sigl et al., 2013; Sigl et al., 2015) and (orange) a stacked Bi record from the South Greenland ice cores (Chellman et al., 2017) . Samples influenced by forest fire and biomass burning (BB) activity were replaced by the corresponding 21-year running median before smoothing the record with an 11-yr filter (thick blue line). BB activity detection is based on co-registered vanillic acid (for D4; McConnell et al., 2007) and on an outlier detection algorithm previously employed for discrimination of fire activity and volcano detection in Greenland ice cores (Fischer et al., 2015; Sigl et al., 2013). b) Stacked rBC_{no BB} record from Greenland and CG03_{no BB} record. Mean rBC_{no BB} concentrations and relative enrichment to Preindustrial (PI) are provided for specific age windows marked by red shading (summarized in Table 3) dominated by coal-burning, petroleum-product burning and present day (PD) emissions.**

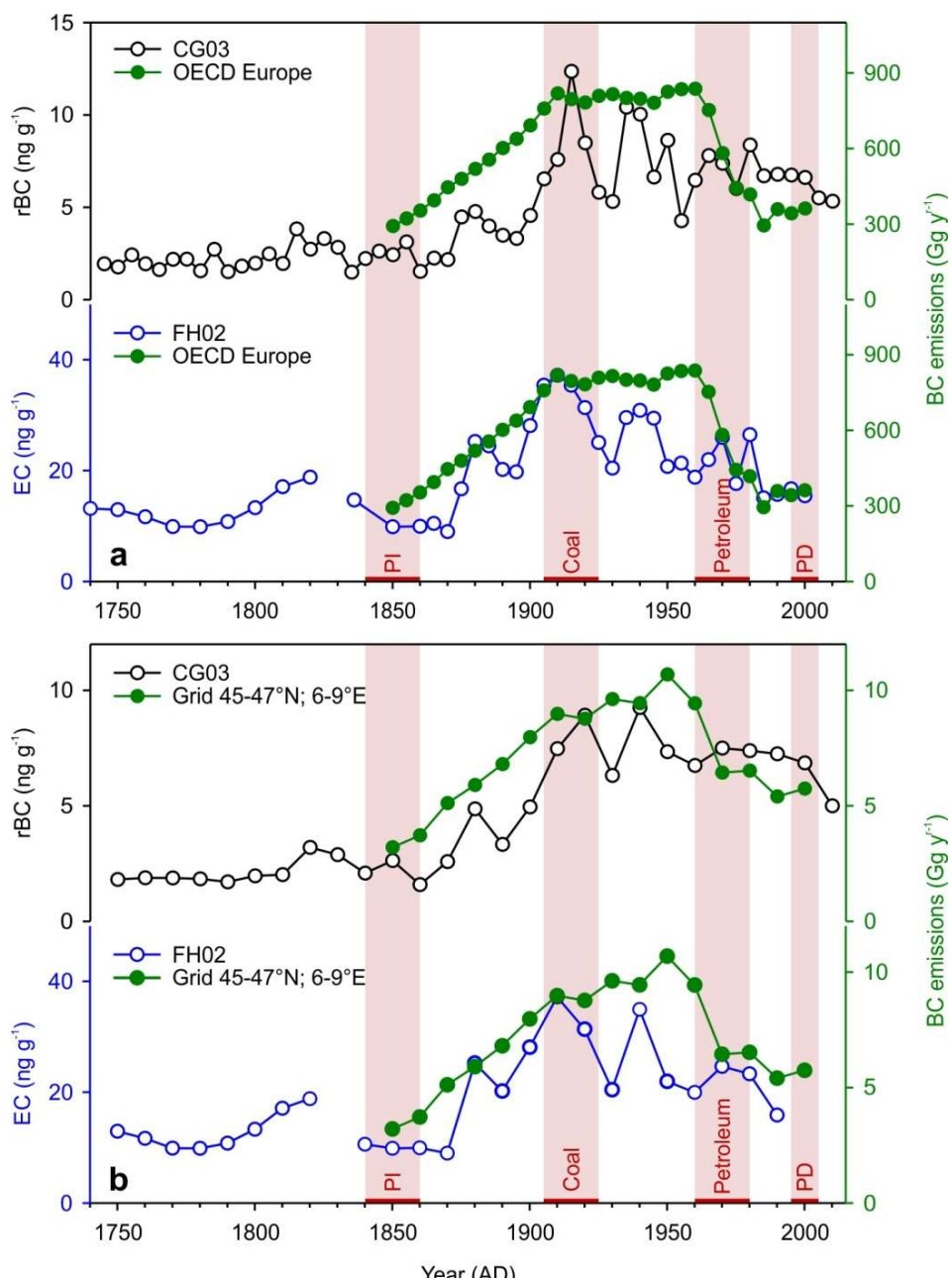

**Figure 6: a) CG03 rBC and FH02 EC concentrations and energy-consumption based BC emission estimates from fossil fuel and biofuel burning (Bond et al., 2007) for OECD Europe and b) for the grid cells 45-47°N and 6-9°E encompassing the locations of the ice-core drilling sites and the four glacier observation records in the Western Alps (see Fig. 1). Mean rBC$_{no\,BB}$ concentrations and relative enrichment to Preindustrial (PI) are provided for specific age windows marked by red shading (summarized in Table 3) dominated by coal-burning, petroleum-product burning and present day (PD) emissions.**

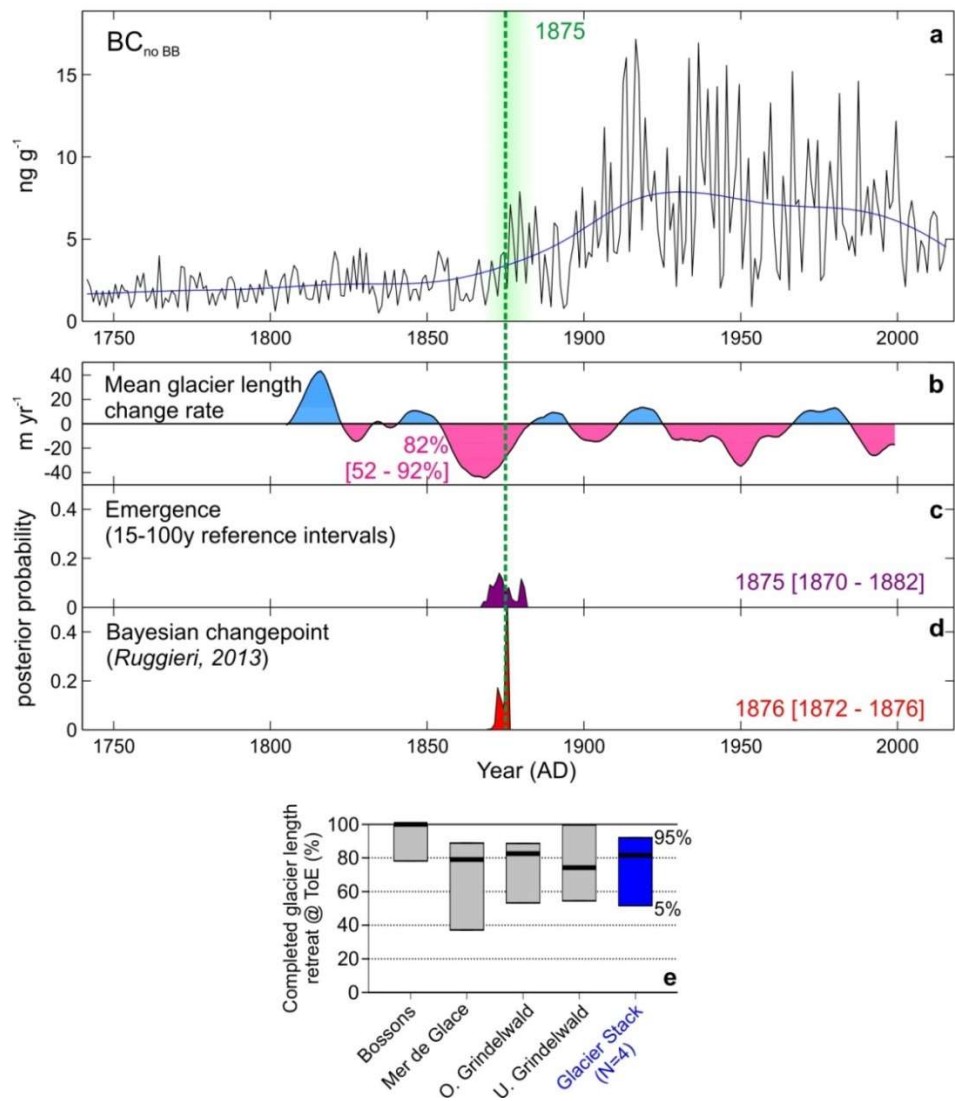

**Figure 7: a)** Annual CG03 non-biomass burning rBC concentrations (BC$_{no\ BB}$, black) with 15-yr filtered trend; **b)** mean glacier length change rate (smoothed with a 11-yr filter) of the stacked (N=4) glacier length records from Bossons, Mer de Glace, Oberer Grindelwald and Unterer Grindelwald glaciers; **c)** normalized distribution of the probabilities of the timing of emergence of industrial BC deposition assessed across 15-100-yr windows using time-of-emergence (ToE) analysis (Hawkins & Sutton 2012) and **d)** using a linear Bayesian change point algorithm (Ruggieri, 2013). The two lower panels give the values for the change points, showing the median [5-95% range], or in the case of the Bayesian change point method the mode [5-95%]. Dashed green line across panels a-d represent the median of the ToE analysis. The read area left of the intersect of the mean glacier length change rate curve (panel b) indicates the completed cumulative length reduction since the 1850s maximum until ToE in 1875 and the 5-95% range taking also into account an absolute dating uncertainty of ±5 years; **e)** completed glacier length reductions since mid-1850s maximum for the four individual glaciers and the stack (black bars are for the year 1875) with 5-95% total uncertainty range.

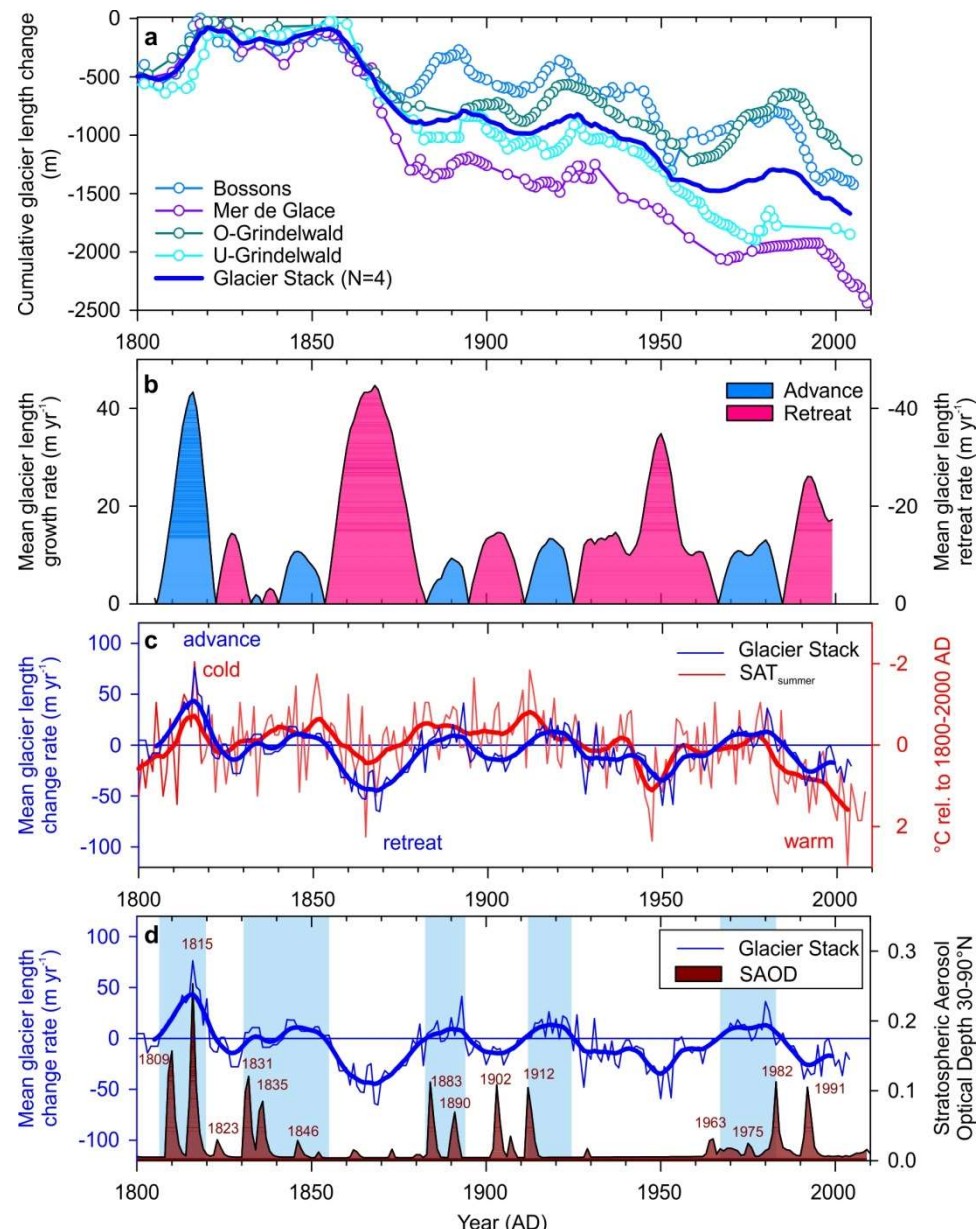

**Fig. 8: a) Cumulative glacier length changes for the four glaciers Bossons, Mer de Glace, Oberer (O-) Grindelwald and Unterer (U-) Grindelwald and their average (*Glacier Stack*; missing observations were filled using linear interpolation); b) mean glacier length change rate (smoothed with a 11-yr filter) of the *Glacier Stack* length record indicating phases of average glacier advances (blue shading) and of glacier retreat (red shading), respectively; c) smoothed and annual resolution mean glacier length change rates of the *Glacier Stack* and equally resolved surface air temperature anomalies for the summer half year (SAT$_{summer}$) from the Greater Alpine Region HISTALP station network (Böhm et al., 2010); d) as c) but with stratospheric aerosol optical depth (SAOD) at 550nm based on ice cores (1800-1850: Toohey & Sigl 2017) merged with the CMIP6 (version 2) reconstruction (1850-2000: Luo 2016, Eyring et al. (2016)).**

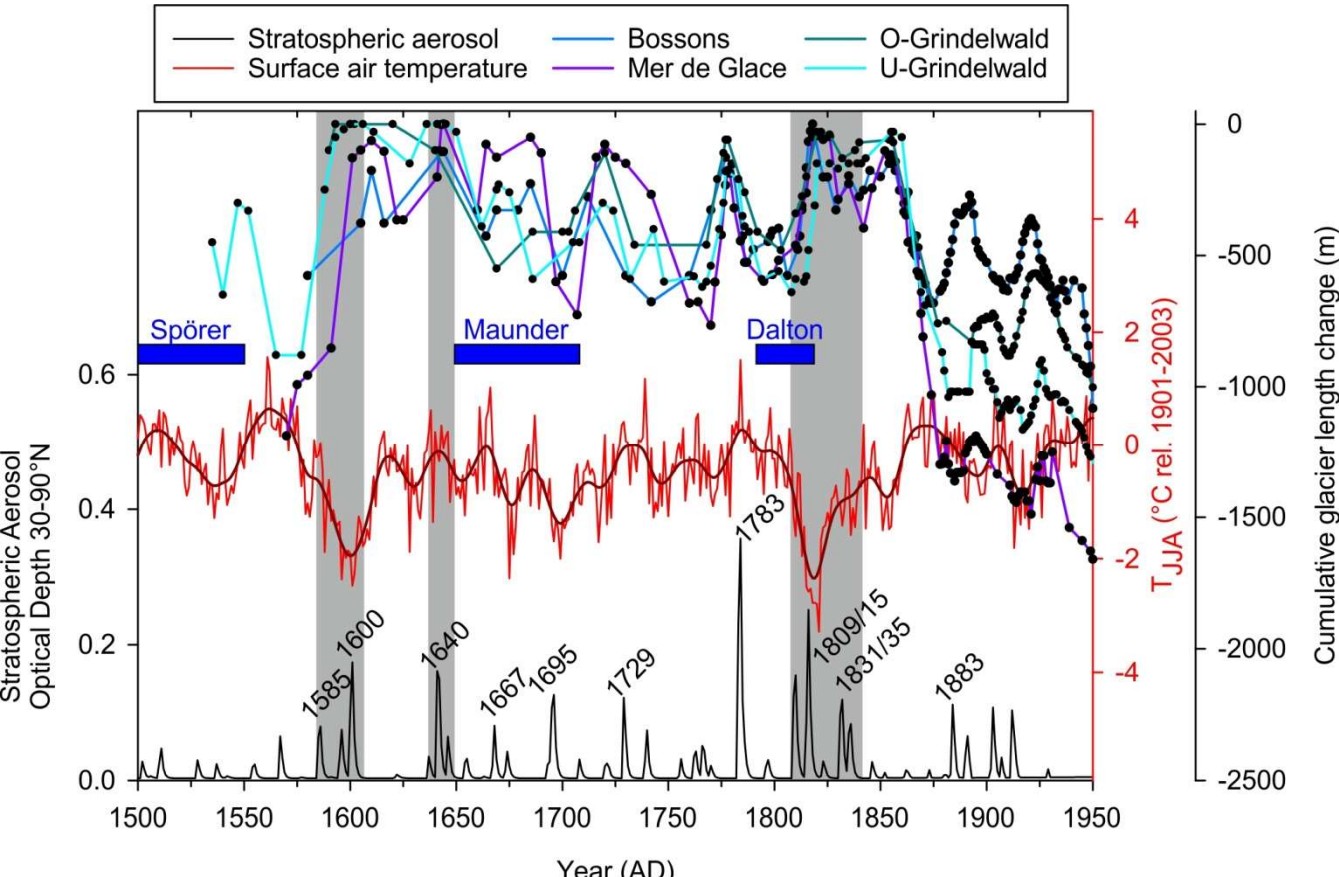

**Fig. 9: Cumulative glacier length changes for the four glaciers Bossons, Mer de Glace, Oberer (O-) Grindelwald and Unterer (U-) Grindelwald with black dots marking years with observations (Nussbaumer & Zumbühl 2012), tree-ring reconstructed Alpine summer (JJA) temperatures (Büntgen et al., 2011), minima in solar activity (Usoskin 2017), and volcanic aerosol forcing (Revell et al., 2017; Toohey and Sigl, 2017) from 1500 to 1950 AD. Grey shading marks time periods with increased volcanic aerosol forcing.**