# Peer review of "19th century glacier retreat in the Alps preceded the emergence of industrial black carbon deposition on high-alpine glaciers"

_The Cryosphere, 2018_

## Referee Comment (RC1) · Anonymous Referee #1 · 23 Mar 2018

This paper sets out to challenge the hypothesis that an increase in industrially-produced black carbon (refractory black carbon; rBC) is responsible for glacier retreat in the Alps during the latter half of the 20th century due to a decreasing albedo feed-back mechanism. The authors use data acquired from 2 new ice cores from the Colle Gnifetti and to derive the timing of an increase in rBC deposition above preindustrial levels and compare it to the timing and rates of glacial retreat, finding that most of the glacier retreat in the Alps had occurred prior to the onset of higher rBC. Another notable finding of the study is a discrepancy in modelled and reported rBC emissions and what is actually reported in the ice cores.

I think that this is an excellent manuscript. The authors do a great job in tying their observations from a single site to observations from other rBC records in the region as

a validation of the regional scope of their conclusions, and using a multi-proxy approach to documenting potential rBC source. The time of emergence analysis is novel for this application and was useful for deriving the emergence of "industrial" rBC as an aerosol.

I thought that Figure 7 was particularly effective because it showed how glacier advance/retreat was functioning independently of rBC concentration where glaciers retreated when rBC concentration is high and advance when it is low.

Minor editorial comments are as follows:

Abstract, line 19: should be "The basis..." Pg 1, line 7: why "already"? Pg 1, line 8: should it be "cloud forming processes"? Pg 1, line 26: "hampering to attribute" is awkward. Pg 4, line 18: why is it summer biased? Pg 7, line 9: missing a period Pg 11, line 26: "forced by the latter" is awckward Pg 12, line 8: "documentaries"? Pg 13, line 11: delete "towards" Pg 14, line 13: maybe change "Much understanding" to "Much of the understanding"?

---

## Short Comment (SC1) · 9 Apr 2018

Commentary by T. H. Painter, M. G. Flanner, G. Kaser, B. Marzeion, R. A. VanCuren, and W. Abdalati

We are quite pleased to see testing of the hypothesis presented in our 2013 paper End of the Little Ice Age in the Alps forced by industrial black carbon. However, the paper submitted here to The Cryosphere Discussions (Sigl et al, hereafter SCD for Sigl Cryosphere Discussion) has a host of logical and interpretive errors that prevent its conclusion that it refutes the hypothesis in Painter et al 2013, let alone robustly. We describe these errors in the following three categories: (1) glaciology, (2) ice core interpretation, and (3) radiative transfer.

[Figure]

(1) Glaciological

The importance and magnitude of the post-1865 retreat of Alpine glaciers is not that they retreat from the LIA high stand of the early 19th century but instead that they retreat to lengths not observed in the previous several hundred years. The Little Ice Age is generally considered as spanning 1300-1870 (Grove, 2004). It is exactly because of this excursion that we used the glacier length records that we did in Painter et al (2013), which reached back hundreds of years. The specific statement in SCD, "can thus be understood as a delayed rebound back to their positions they had before the radiative perturbed time period 1800-1840 AD" (p12, lines 20-21) is inconsistent with the glacier records from the Alps going back to the 1600s. Such an abrupt Alps-wide excursion from multi-centennial glacier length equilibrium range requires a likewise abrupt and marked perturbation to energy or mass balance (Huybrechts et al., 1989; Kerschner, 1997) or potentially an intensification of subannual climate variability (Farinotti, 2013). Such do not exist in the observational record nor the HISTALP reconstruction. The excursion from the envelope of lengths did not happen until approximately 1870-1875, consistent with their interpretation of when BC emissions emerged above pre-industrial.

Note that in Painter et al (2013), we state on p2, first paragraph "As indicated by Huybrechts et al. (10) and many others, our best understanding of changes in temperature and precipitation in the 19th century indicates that there was no regional climatic anomaly coincident with the coherent retreat of glaciers in the Alps near 1865". Moreover, in our Figure 1 (Painter et al 2013), the vertical dashed line at 1875 indicates the unambiguous excursion in lengths from the previous several hundred years (Figure 1 below). We understand the appeal of the higher temporal resolution glacier length records but without the records back into several earlier centuries, one cannot address the driver of that excursion and the relevant scientific question.

An additional issue with the current document is that SCD do not address the explicit treatment of glacier mass balance in our paper. With the temperature and precipitation

from HISTALP, the glacier mass balance model matches well the record of the Hintere-isferner across the last ~60 years. However, the excursion in glacier length cannot be resolved without contribution of some additional forcing, as Huybrechts et al (1989) put it, "Forcing the mass balance history [with summer and mean annual temperature anomalies] brought to light, that, in particular, the observed glacier retreat since about 1850 is not fully understood. This result and the improved model simulations that could be obtained while assuming an additional negative mass balance perturbation during roughly the last 150 years, seems to point to additional features affecting the glacier's mass balance that are not captured well in the ambient climatic records.

SCD seem to allude to some contribution from the temperatures from Buntgen et al. (2006; 2011) as being suggestive of the rising temperatures sufficient to explain the post-1865 glacier length excursion. However, these reconstructions are inconsistent with the observational record, even after it is corrected for shading.

SCD cite Gabbi et al (2015) multiple times but miss the contradictory relevance of the central points of that paper compared to their paper from a glaciological, ice core, and radiative transfer standpoint. In particular, at the Clarindenfirn, across most of the 20th century BC has ~3X greater radiative forcing than dust, and summer radiative forcings from BC in snow were 13-16 W m-2. Gabbi et al consider this a lower limit, including this being a period of markedly lower BC emissions, deposition, and radiative forcing in snow than at the peaks of industrialization. Considering the SCD BC time series, present day radiative forcings should be relatively consistent with those in the 1880s, before the further rise in the late 1800s to that of the first decades of the 20th century. Indeed, they indicate "As a result, we obtain similar BC concentrations in the surface layer on average, and thus, a comparable impact of BC on glacier mass balance. The general agreement of our assessment with that of Painter et al. (2007) [our dust radiative forcing and melt paper] indicates the highly relevant role of BC in shaping changes in glacier mass balance over the last century."

(2) Ice Core Interpretation

In the context of ice core interpretation, this is the area of the authors' expertise. We in no way contest their analysis of the constituents that were found in the CG ice core. The issue comes with the interpretation of the results. Interpretation of poorly mixed constituents (such as BC) in complex terrain with enormous relief must be treated with caution and, as such, semi-quantitatively. In particular, ice cores from locations with such intense wind scouring and complex wind fields (Figure 2) cannot be considered to be absolute in capturing all air that has passed over and onto the surface. Moreover, their flow regimes are frequently disconnected from those in the ablation zones (as they were for the mid 19th century glaciers at 1500 to 2200 m elevation) where the increased net fluxes from impurities would have had their maximum impact on mass balance and glacier length. As such, unlike determination of well-mixed gases in ice cores, these cores should be considered as suggestive of transport to mountain systems and offering a lower bound on deposition.

Multiple sources suggest that BC emissions from industrialization on the European continent began to increase substantially in the 1850s with a quasi-monotonic growth in emissions (Bond et al., 2007; Bond et al., 2013; Lavanchy et al., 1999). SCD argue that, because such reconstructions have substantial uncertainties, that the quasi-monotonic increase must be erroneous and that the time series in the ice core more accurately represents actual emissions. This argument is posited with no argument to explain 100% excursions in BC emissions that are expressed in the ice core time series presented in SCD Figure 3a, when BC drops entirely (annual) or by ~50% (5 yr) from 1880 to 1890. Likewise, no explanation is given as to the steep drop from ~1920 and then the enormous climb beginning in 1930, coincident with the global depression. The timing of these excursions is markedly out of phase with well understood historical excursions in continental European productivity.

Regarding the non-quantitation between the core and regional emissions, the inter-annual variation in the laser-incandescence (LI) data is quite large. SCD contends that the linear interpolation between years in the Bond et al emission estimates is not

credible. Here, we show European emissions data for 2000-2015 (Figure 3), which span the largest economic variation in more than half a century. Note that PM2.5 emissions still had an average interannual variation of 3 percent. This suggests that the noise in the ice core BC concentrations is in transport variation year to year, not to first order emissions. The burden needs to be put back to the authors to defend the "emergence" statistic. Indeed, the more noise, the later the lag in detection of any increase in LI-sensitive carbon. These thoughts are detailed in the four points below:

1. The use of summer-only data assumes that summer aerosols at the Col represent the operative aerosol deposition impacting the whole glacier - the strong persistent inversion in the Alps in summer disconnects the atmosphere at the Col from the atmosphere over the transport and ablation zones of the valley glaciers (Figure 4), thus the Col record is not a quantitative analog of valley conditions, and there is no evidence presented that winter and spring deposition can be inferred from the summer data. 2. The laser incandescence (LI) method is highly sensitive to carbonaceous chain aggregate particles and is usually calibrated with a pure hydrocarbon combustion soot generator. The measurement is less sensitive to "brown carbon" and mixed-component aerosols such as produced by burning low quality coal or inefficient coal combustion (Sun et al., 2017). It may be that the record reflects evolving combustion technology as much or more so than emission strength. 3. The large inter-annual variation in the LI data is inconsistent with interpreting it as a surrogate for regional or continental scale emissions. Data on modern fine particle emissions from 2000 to 2015 (Figure 3) show an average inter-annual variation of about 3%, even though this period includes the enormous economic fluctuations driven by the crisis of 2008. This suggests a large variability in the LI data due to meteorology. Hence, these data suggest noise is in transport variation year to year, not emissions. 4. The emergence test statistic is strongly influenced by the noise in the time series data - with detection of a trend delayed as noise increases. Since the noise in these data contains a large meteorological component, the inferred timing of the beginning of industrial impact in these data has a large and un-discussed time lag.

Note also that in SCD Figure 5, the climb in BC in Greenland unambiguously occurs around 1850-1860.

(3) Radiative Transfer

SCD do not address the magnitude of radiative forcing by BC that was present in the Alps at the time. For the claim of no role for BC in the retreat of European glaciers from the LIA to be valid, the radiative forcings in Painter et al (2013) estimated from the ice core BC would have to be overestimated by more than an order of magnitude. We think this is highly unlikely given the known increase in aerosol concentrations with decreasing altitude, as substantiated by SCD co-author Schwikowski (2004), among many others. From an energy balance perspective, we do not understand how SCD would explain away the 20-40 W m-2 seasonally-averaged radiative forcing by BC for April-June in the ablation zones? As described in Painter et al (2013), the melt magnitude associated with the 1880 radiative forcing of 10-20 W m-2 would have been 240-480 kg m-2 (0.6-1.2 m w.e.) and with the 1900 radiative forcing of 19-38 W m-2 would have been 450-890 kg m-2 (1.1-2.2 m w.e.). Equivalent changes in temperature to produce such radiative forcings in context of mass balance would have reached 3-4 K.

SCD state that our study suffers from not including the radiative forcing by dust in snow (p3, lines 32-35+p4, 1-2). To a degree they are correct because we did not include the description of our coupled dust+BC radiative forcings. However, the BC radiative forcings that we report were in fact (BC+Dust)-(Dust Only). Moreover, we concluded from extensive analyses of these cores that dust deposition saw no trends during the period recorded in the ice cores. Likewise, SCD also found that mineral dust had no trends (p8, 30-32), "In agreement with other dust records from Colle Gnifetti (Bohleber et al., 2018; Wagenbach and Geis, 1989), we observe no enhanced mean (or frequency) of mineral dust deposition throughout the 19th century (Supplementary Figs. S1, S4)." Either way, we are puzzled that such would be highlighted as a 'suffering'.

[Figure]

**Summary**

Again, we are enthused that our work has stimulated thought and testing of our hypothesis. However, based on the above comments, it appears to be clear that SCD does not refute the hypothesis. Ultimately, the core of the SCD paper is really to reconcile an issue with ice cores and emission scenarios, neither of which presently can be considered quantitatively robust in characterizing regional atmospheric conditions at the elevations of Alpine glacier ablation zones (today back to the high stand of the LIA). Instead, the ice cores can be considered as suggestive of BC deposition timing and magnitude but not as an absolute quantification. Likewise, to a lesser degree the paper also encounters the discrepancy between the temperature observations, HISTALP reconstructions, and the tree-ring derived temperature reconstruction of Büntgen et al (2012). Let's work as a community to resolve these particular discrepancies.

References:

Bond, T. C., Bhardwaj, E., Dong, R., Jogani, R., Jung, S., Roden, C., . . . Trautmann, N. M. (2007). Historical emissions of black and organic carbon aerosols from energy-related combustion, 1850-2000. Global Biogeochemical Cycles, 21(2), GB2018. doi:10.1029/2006GB002840

Bond, T. C., Doherty, S. J., Fahey, D. W., Forster, P. M., Berntsen, T., DeAngelo, B. J., . . . Zender, C. S. (2013). Bounding the role of black carbon in the climate system: A scientific assessment. J. Geophys. Res. doi:10.1002/jgrd.50171

Farinotti, D. (2013). On the effect of short-term climate variability on mountain glaciers: insights from a case study. J. Glaciol., 59(217), 992-1006. doi:10.3189/2013JoG13J080

Gabbi, J., Huss, M., Bauder, A., Cao, F., & Schwikowski, M. (2015). The impact of Saharan dust and black carbon on albedo and long-term glacier mass balance. The Cryosphere, 9, 1133-1175. doi:10.5194/tcd-9-1133-2015

Grove, J. M. (2004). Little ice ages: ancient and modern (Vol. 1 and 2). London: Routledge.

Huybrechts, P., Nooze, P. d., & Decleir, H. (1989). Numerical modeling of Glacier d'Argentiere and its historical front variations. In J. Oerlemans (Ed.), Glacier Fluctuations and Climatic Change (pp. 373-389): Kluwer Academic Publishers.

Kerschner, H. (1997). Statistical modelling of equilibrium-line altitudes of Hintereisferner, central Alps, Austria, 1859-present. Ann. Glaciol., 24, 111-115.

Lavanchy, V. M. H., Gaggeler, H. W., Schotterer, U., Schwikowski, M., & Baltensperger, U. (1999). Historical record of carbonaceous particle concentrations from a European high-alpine glacier (Colle Gnifetti, Switzerland). J. Geophys. Res., 104(D17), 21,227-221,236. doi:10.1029/1999JD900408

Painter, T. H., Flanner, M., Marzeion, B., Kaser, G., VanCuren, R., & Abdalati, W. (2013). End of the Little Ice Age in the Alps forced by black carbon. Proc. Nat. Acad. Sci. USA. doi:10.1073/pnas.1302570110

Schwikowski, M. (2004). Reconstruction of European air pollution from alpine ice cores. In L. D. Cecil (Ed.), Earth Paleoenvironments: Records Preserved in Mid- and Low-Latitude Glaciers (pp. 95-119). Netherlands: Kluwer Academic.

Sun, J., Zhi, G., Hitzenberger, R., Chen, Y., Tian, C., Zhang, Y., . . . Mo, Y. (2017). Emission factors and light absorption properties of brown carbon from household coal combustion in China. Atmospheric Chemistry and Physics, 17, 4769-4780. doi:10.5194/acp-17-4769-2017

[Figure]

[Figure]

**Fig. 1.** From Painter et al (2013), their Figure 1 shows the vertical line plotted at 1875 to indicate the unambiguous excursion from the previous several hundred years of glacier length.

[Figure]

**Fig. 2.** Photograph of frequent wind structures in snow at the Colle Gnifetti, demonstrating that the site is subject to frequent redistribution of snow and its constituents. (Source: U. of Maine)

**Fig. 3.** European emissions of SOx, NOx, NMVOC, and PM2.5 across 2000-2015, which includes the global economic downturn. Data source: European Environment Agency.

[Figure]

**Fig. 4.** Summer looking south into the Bernese Alps shows air pollution in the Alps confined to lower altitudes, concentrating the deposition of soot and dust on the lower slopes. Credit: Peter More

---

## Referee Comment (RC2) · Anonymous Referee #2 · 29 Apr 2018

This paper evaluates the hypothesis that black carbon deposition could play a dominant role in European glacier dynamics. The paper presents black carbon concentrations found in ice cores at Colle Gnifetti along with other tracers of different kinds of combustion. Authors examine the role of mineral dust in possible forcing; since it absorbs light, mineral dust can be a possible confounding factor. They also compare their measurements with those of other ice cores. Finally, they compare the trends of black carbon deposition with those of black carbon emission and point out discrepancies between measured tracers and bottom-up inventories. The work is supported by a careful treatment of timing and uncertainties to interpret the ice core measurements.

Overall, this part of the paper is a quite thorough and welcome contribution to the discussion of black carbon (and other species) emissions and influence during the

industrial era. I commend the authors on their careful work.

The next part of the work– and the origin of the paper's title– compares the timing of glacial retreat with the timing of black carbon increase. These glaciers are frequently observed, making them good candidates for such an analysis. Authors identify retreat and advance periods and compare them with time-of-emergence of black carbon above pre-industrial periods, finding no relationship. Finally, on page 11, authors posit that volcanic forcing, and not change in albedo caused by black carbon deposition, is the cause of glacial retreat. The glaciers analyzed (four in a "glacier stack") begin to retreat before the increase in black carbon emissions, so it is unlikely that black carbon, alone, caused the current retreat.

Despite this good point, this part of the work appears less supported by the evidence presented. An important question is on what time and spatial scales one respects a response between a forcing and a regional response. Attribution of climate response typically involves some kind of large-scale pattern matching, considering more factors than given here. One doesn't expect an increase in black carbon emission to correlate neatly with the glacier retreat– although certainly the fact that glaciers retreated first indicates that other causes are at work. If there were such neat correlations, we should have much less trouble identifying the causes of climate change, overall. So the following questions would have to be answered in order to confidently state the "No role for black carbon" as in the title: What other factors could contribute to glacial retreat; How much do they vary and on what temporal and spatial scales (i.e. what noise could confound the signal and must be averaged out); and *then* how much black carbon does contribute and whether it has a significant effect.

The authors also considered volcanic forcing, which they suggest to be much more relevant than black carbon forcing, yet they did not provide any quantification of or data behind the volcanic forcing, but only some discussion. That quantification would be needed in order to make the statements in this paper with confidence.
This paper is well-written and organized. One editorial comment: page 3, line 24 should read "light absorbing" instead of "light adsorbing."

In summary, I wholeheartedly support publication of the quite careful work on reconstruction of the black carbon and tracer deposition in ice, the comparison with previous ice cores and the discussion of the mismatches with emission inventories. This is a worthy contribution in itself and it could be published without modification.

In order to publish the discussion on the connection to forcing (and, it seems, the intent/title of the paper), there should be a broader discussion of the spatial and temporal patterns, variability, and other causes. That seems like a substantial amount of effort and I do not wish to minimize the authors' excellent contribution here. I suggest that perhaps, in order to proceed quickly toward publication, authors could include the timing of glacier retreat, point out that many other factors are at play including the volcanic forcing, and soften the title and statements regarding "no role" until a fuller analysis is done. I shouldn't be surprised if authors are already engaging in such a broader analysis.

---

## Author Comment (AC1) · 17 May 2018

We thank the referees and colleagues for their time and energy to review our manuscript. Please find below our responses to the points and comments raised by the reviewers and peers and the changes in the manuscript.
* * *
**Reviewer #1 (RC1):**

This paper sets out to challenge the hypothesis that an increase in industrially produced black carbon (refractory black carbon; rBC) is responsible for glacier retreat in the Alps during the latter half of the 20th century due to a decreasing albedo feedback mechanism. The authors use data acquired from 2 new ice cores from the Colle Gnifetti and to derive the timing of an increase in rBC deposition above preindustrial levels and compare it to the timing and rates of glacial retreat, finding that most of the glacier retreat in the Alps had occurred prior to the onset of higher rBC. Another notable finding of the study is a discrepancy in modelled and reported rBC emissions and what is actually reported in the ice cores.

I think that this is an excellent manuscript. The authors do a great job in tying their observations from a single site to observations from other rBC records in the region as a validation of the regional scope of their conclusions and using a multi-proxy approach to documenting potential rBC source. The time of emergence analysis is novel for this application and was useful for deriving the emergence of "industrial" rBC as an aerosol. I thought that Figure 7 was particularly effective because it showed how glacier advance/retreat was functioning independently of rBC concentration where glaciers retreated when rBC concentration is high and advance when it is low.

Minor editorial comments are as follows:
Abstract, line 19: should be "The basis. . ." Pg 1, line 7: why "already"? Pg 1, line 8: should it be "cloud forming processes"? Pg 1, line 26: "hampering to attribute" is awkward., Pg 7, line 9: missing a period Pg 11, line 26: "forced by the latter" is awkward Pg 12, line 8: "documentaries"? Pg 13, line11: delete "towards" Pg 14, line 13: maybe change "Much understanding" to "Much of the understanding"?

We kindly acknowledge this positive and constructive evaluation. We carefully considered all the suggested technical corrections and revised the manuscript accordingly.

Pg 4, line 18: why is it summer biased?

Due to preferential wind erosion of winter snow as stated in line 15 and in the cited references

**Short Comment by T. H. Painter, M. G. Flanner, G. Kaser, B. Marzeion, R. A. VanCuren, and W. Abdalatiet al (SC1):**

We are quite pleased to see testing of the hypothesis presented in our 2013 paper End of the Little Ice Age in the Alps forced by industrial black carbon. However, the paper submitted here to The Cryosphere Discussions (Sigl et al, hereafter SCD for Sigl Cryosphere Discussion) has a host of logical and interpretive errors that prevent its conclusion that it refutes the hypothesis in Painter et al 2013, let alone robustly. We describe these errors in the following three categories: (1) glaciology, (2) ice core interpretation, and (3) radiative transfer.

We acknowledge the comments by the authors of Painter et al., 2013; PNAS (hereafter PP13) and, in the following, address them point-by-point. Our main conclusions stand untouched.
1) At the time when BC deposition values from multiple state-of-the-art ice-core reconstructions exceeded its natural variability, the vast majority of glacier length reductions had already occurred.
2) During 1910-1920, when industrial BC deposition reached peak values exceeding three times values from 1875-1885 AD, many glaciers in the Alps were advancing.
Both these observations are inconsistent with the hypothesis that industrial BC forced the end of the Little Ice Age in the Alps. The end of the Little Ice Age, we argue, was primarily due to the absence of the radiative forcing agents that produced the Little Ice Age in the first place (Miller et al., 2012; Schurer et al., 2014) – clusters of large volcanic eruptions and solar minima, both more abundant between 1600-1840 AD compared to the long-term mean (Toohey and Sigl, 2017; Usoskin, 2017).

(1) Glaciological
The importance and magnitude of the post-1865 retreat of Alpine glaciers is not that they retreat from the LIA high stand of the early 19th century but instead that they retreat to lengths not observed in the previous several hundred years.
The Little Ice Age is generally considered as spanning 1300-1870 (Grove, 2004). It is exactly because of this excursion that we used the glacier length records that we did in Painter et al (2013), which reached back hundreds of years. The specific statement in SCD, "can thus be understood as a delayed rebound back to their positions they had before the radiative perturbed time period 1800-1840 AD" (p12, lines 20-21) is inconsistent with the glacier records from the Alps going back to the 1600s. Such an abrupt Alps-wide excursion from multi-centennial glacier length equilibrium range requires a likewise abrupt and marked perturbation to energy or mass balance (Huybrechts et al., 1989; Kerschner, 1997) or potentially an intensification of subannual climate variability (Farinotti, 2013). Such do not exist in the observational record nor the HISTALP reconstruction.

We argue that the glaciers were not within their equilibrium range, neither from 1800-1840 nor from 1600-1840 due to the repeated radiative perturbations

during the late LIA. The glacier length records Mer de Glace and Bossons provide both: 1) a long-term perspective extending before 1600 AD and 2) high-resolution observations since the mid-1850s that allow to identify precisely the timing of increased glacier retreat rates (Nussbaumer and Zumbühl, 2012; Zumbühl et al., 2008). Glacier lengths of Bossons were equal to that of 1870 in 1580, 1700, 1730 and 1760 AD (see Figure below). In 1742 AD the length was less than at any time between 1880-1930 AD. The glacier length positions of Mer de Glace during the late 16th century had not been reached before 1878 AD. The 16th and 18th century are characterized by lower volcanic activity, whereas the early 17th and early 19th century experienced strong volcanic perturbations. Mean Stratospheric Aerosol Optical Depth for the time window 1600-1840 AD was 40% larger than during the entire Common Era (Sigl & Toohey, 2017). Long-term glacier length records (e.g. from Great Aletsch and Gorner Glacier; (Holzhauser et al., 2005)) clearly show that glacier lengths less than in the 1880 AD have frequently occurred throughout the Common Era (e.g., in the 16th and 14th century and during most of the 'Medieval Quiet Period' and 'Roman Quiet Period', without the presence of industrial BC. See also review paper by Solomina et al. (2016).

*This aspect is now considered in the "Discussion" section of the revised manuscript, where the following sentence was added: "The stratospheric aerosol burden for the time window 1600-1840 AD was 40% larger than during the entire Common Era (Sigl & Toohey, 2017) with volcanic eruptions frequently forcing cold spells and glacier advances (e.g., in 1600s, 1640s, 1820s, 1840s) in the Alps (Fig. 9) and elsewhere (Solomina et al., 2016)."*

[Figure]

*Figure 9: Cumulative length changes of four Alpine glaciers (Nussbaumer & Zumbühl 2012), tree-ring reconstructed Alpine summer temperatures (Büntgen et al., 2011), minima in solar activity (Usoskin 2017) and volcanic aerosol forcing (Revell et al., 2017; Toohey and Sigl, 2017) from 1500 to 1950 AD. Grey shading marks time periods with increased volcanic aerosol forcing.*

The excursion from the envelope of lengths did not happen until approximately 1870-1875, consistent with their interpretation of when BC emissions emerged above pre-industrial.

It appears the authors here suggest that the Alpine glacier retreat (accelerating in the 1860s, based on high-resolution glacier length records) was well within the range of natural variability until 1875 AD (keeping in mind that climate was generally very cold since 1600 AD). We agree. The glacier retreat until 1875, however, makes up over 80% of the entire 1850-1900 length reduction (see Table below) visualizing the "*End of the Little Ice Age*" (PP13). For the remaining 19th century we stated: "*Only after 1870 AD, when BC emissions started to strongly increase, snow-albedo impurity effects may have potentially contributed to the glacier length reductions, although the magnitude of such a feedback must be considered small given that glaciers were advancing during the coal-burning era of peak air pollution with BC in Central Europe (1910-1920)*"; p.14, l.9).

In the revised manuscript, following suggestions of Reviewer #2 we changed the title to "*No **leading** role for industrial black carbon in forcing 19th century glacier retreat in the Alps*".

*Table: Glacier length variability, and BC concentrations in the Alps (Colle Gnifetti) and Greenland (4 ice-core stack) during the mid-19th to early 20th century.*

| Year | BC$_{Alps}$ (ng/g) | BC$_{Greenland}$ (ng/g) | Bossons (m) | Mer de Glace (m) | O. Grindelwald (m) | U. Grindelwald (m) |
|------|------|------|------|------|------|------|
| 1850 | 2.5 | 1.4 | -200 | -149 | -63 | -72 |
| 1875 | 4.1 | 1.6 | -682 | -1099 | -696 | -780 |
| 1900 | 4.9 | 3.4 | -527 | -1261 | -734 | -950 |
| 1915 | 10.8 | 5.1 | -549 | -1419 | -777 | -1070 |

*ice core concentrations are 10-yr means (e.g., 1846-1855 AD)

Note that in Painter et al (2013), we state on p2, first paragraph "*As indicated by Huybrechts et al. (10) and many others, our best understanding of changes in temperature and precipitation in the 19th century indicates that there was no regional climatic anomaly coincident with the coherent retreat of glaciers in the Alps near 1865*". Moreover, in our Figure 1 (Painter et al 2013), the vertical dashed line at 1875 indicates the unambiguous excursion in lengths from the previous several hundred years (Figure 1 below). We understand the appeal of the higher temporal resolution glacier length records but without the records back into several earlier centuries, one cannot address the driver of that excursion and the relevant scientific question.

Using the glacier records going back several hundreds of years the dominant role of volcanic eruptions in driving summer temperature variability and thus glacier excursions becomes even more evident. Our new Figure 9 shows that all major glacier advances since 1600 AD (e.g., in 1600s, 1640s, 1820s and 1850s) followed closely clusters of large stratospheric eruptions which has also been pointed out in several other recent studies (Lüthi, 2014; Solomina et al., 2016). The Huybrechts study from 1989 was long before the "*instrumental warm bias*"

entered the scientific discussion (Böhm et al., 2010; Frank et al., 2007) and also dates back to two years prior to the eruption of Pinatubo 1991 when the global-scale cooling impact of volcanic eruptions was demonstrated (Robock and Mao, 1995).

An additional issue with the current document is that SCD do not address the explicit treatment of glacier mass balance in our paper. With the temperature and precipitation from HISTALP, the glacier mass balance model matches well the record of the Hintereisferner across the last ~60 years. However, the excursion in glacier length cannot be resolved without contribution of some additional forcing, as Huybrechts et al (1989) put it, "Forcing the mass balance history [with summer and mean annual temperature anomalies] brought to light, that, in particular, the observed glacier retreat since about 1850 is not fully understood. This result and the improved model simulations that could be obtained while assuming an additional negative mass balance perturbation during roughly the last 150 years, seems to point to additional features affecting the glacier's mass balance that are not captured well in the ambient climatic records.

SCD seem to allude to some contribution from the temperatures from Büntgen et al. (2006; 2011) as being suggestive of the rising temperatures sufficient to explain the post-1865 glacier length excursion. However, these reconstructions are inconsistent with the observational record, even after it is corrected for shading.

Since the work from Huybrechts et al. (1989), almost thirty years ago, a number of new studies have focused on the issue of an early 19th century instrumental warm bias producing an apparent cooling trend in the 19th century (see PP13 Fig. 2). Early instrumental station measurements prior to the 19th century were specifically prone to error and scarce in high-alpine environments. Correcting this bias and reconstructing early 19th century temperatures and precipitation rates in high-alpine regions thus remain areas of ongoing research (Böhm et al., 2010; Frank et al., 2007). Biases due to unshaded temperature readings are difficult to correct and likely result in an overall net warming influence on early station data (Böhm et al., 2010).

A recent multi-proxy reconstruction shows a mean 19th century JJA warming trend for Europe of 0.4°C (Luterbacher et al., 2016). The Alpine-wide tree-ring reconstruction from Büntgen et al., (2011) shows a 19th century warming trend of 1.5°C. With no apparent 19th century cooling trend in state-of-the-art Alpine climate reconstructions, there is no need to invoke additional feedbacks to produce glacier retreat. The fact that modelling with HISTALP data give reasonable agreement with observations during the past 60 years, does not mean that HISTALP data must be correct for the early 19th century (Böhm et al., 2010).

Finally, Lüthi (2014), by using a macroscopic model of glacier dynamics, derived a history of glacier equilibrium line altitude changes since 400 AD that strongly resembles reconstructions of Alpine summer temperatures. He concluded that *"the glacier advances during the LIA, and the retreat after 1860, can thus be mainly attributed to temperature and volcanic radiative cooling."*

SCD cite Gabbi et al (2015) multiple times but miss the contradictory relevance of the central points of that paper compared to their paper from a glaciological, ice core, and radiative transfer standpoint. In particular, at the Clarindenfirn, across most of the 20th century BC has ~3X greater radiative forcing than dust, and summer radiative forcings from BC in snow were 13-16 W m-2. Gabbi et al consider this a lower limit, including this being a period of markedly lower BC emissions, deposition, and radiative forcing in snow than at the peaks of industrialization. Considering the SCD BC time series, present day radiative forcings should be relatively consistent with those in the 1880s, before the further rise in the late 1800s to that of the first decades of the 20$^{th}$ century. Indeed, they indicate "As a result, we obtain similar BC concentrations in the surface layer on average, and thus, a comparable impact of BC on glacier mass balance. The general agreement of our assessment with that of Painter et al. (2007) [our dust radiative forcing and melt paper] indicates the highly relevant role of BC in shaping changes in glacier mass balance over the last century."

We cite Gabbi et al (2015), because, in this study, the annual BC record from Fiescherhorn was for the first time reported. The Gabbi et al. study showed that in the 20th century, the peak period of the industrialization in Central Europe with the highest BC concentrations in the Fiescherhorn ice core, annual melt was amplified by maximal 19% due to the combined effect of BC and mineral dust. Thus, the effect must have been much lower in the mid-to late 19th century.

(2) Ice Core Interpretation
In the context of ice core interpretation, this is the area of the authors' expertise. We in no way contest their analysis of the constituents that were found in the CG ice core.

The issue comes with the interpretation of the results. Interpretation of poorly mixed constituents (such as BC) in complex terrain with enormous relief must be treated with caution and, as such, semi-quantitatively. In particular, ice cores from locations with such intense wind scouring and complex wind fields (Figure 2) cannot be considered to be absolute in capturing all air that has passed over and onto the surface.

We acknowledge (and have acknowledged throughout the manuscript) the inherent limitations arising from transport and snow conservation on the mean aerosol concentrations on small temporal and spatial scales. Our main conclusions are, however, drawn from long-term trends ("emergence") that are observed at multiple different sites (two from the Alps, four from Greenland) using multiple different methods and proxies (e.g. BC, EC, Bi) for industrial pollution. The consistency of results across multiple sites indicates that local and site-specific deposition/preservation factors have not substantially altered the long-term signals of black carbon in these ice cores.

Moreover, their flow regimes are frequently disconnected from those in the ablation zones (as they were for the mid-19th century glaciers at 1500 to 2200 m elevation) where the increased net fluxes from impurities would have had their maximum impact on mass balance and glacier length. As such, unlike

determination of well-mixed gases in ice cores, these cores should be considered as suggestive of transport to mountain systems and offering a lower bound on deposition.

The glacier sites from which the ice-cores were drilled are almost certainly more frequently above the planetary boundary layer, but they are by no means decoupled from the lower elevations where the emissions take place. The diurnal cycle as well as the annual cycles of aerosol deposition resulting from convective transport have been tracked and analysed at Colle Gnifetti and Jungfraujoch over many years using remote sensing and high-resolution in-situ aerosol monitoring (Lugauer et al., 1998; Nyeki et al., 2000). Sulphate records from Colle Gnifetti also closely mirror emission estimates over the past 100 years (Engardt et al., 2017). Ice cores from Colle Gnifetti are, in fact, dated by counting the summer peaks of maximum aerosol concentrations for many hundred years back in time (Bohleber et al., 2018). Aerosols from mining emissions during Roman times have been detected in two ice cores as far away as Greenland (Hong et al., 1994; McConnell et al., 2018). In other words, if there were any significant industrial BC emission sources from Central Europe present during the 1850-60s, they would have left detectable traces in the two ice cores from the Alps as well as in the ice cores from Greenland.

Multiple sources suggest that BC emissions from industrialization on the European continent began to increase substantially in the 1850s with a quasi-monotonic growth in emissions (Bond et al., 2007; Bond et al., 2013; Lavanchy et al., 1999).

This statement is hardly true. Bond et al. (2013) is a review paper. The Bond et al., (2007) paper discusses the development of a global emission inventory. Therein, Table 1 lists a single primary data source prior to 1923 AD – fossil fuel production and consumption estimates derived from Andres et al. (1999). We reviewed all these papers and we discuss in our manuscript, that there is scarcely any primary input data available from Central Europe and thus BC estimates from these inventories closely resemble energy consumption estimates taken from countries with more abundant data (i.e., the U.S.), a caveat already discussed by Bond et al. (2007).

Lavanchy (1999), in the first attempt to quantify BC and EC in Colle Gnifetti ice cores, defined the preindustrial as 1740-1889 AD (see Figure below) which is in close agreement with the best estimates given for Greenland for a "*rapid increase in ~1888*" (McConnell et al., 2007) and in many subsequent studies that we have cited in our manuscript.

[Figure]

*Figure. Minimum and maximum as well as 95th, 75th, 50th, 25th, and 5th percentiles of BC and EC from Colle Gnifetti (from Lavanchy et al., 1999).*

SCD argue that, because such reconstructions have substantial uncertainties, that the quasimonotonic increase must be erroneous and that the time series in the ice core more accurately represents actual emissions. This argument is posited with no argument to explain 100% excursions in BC emissions that are expressed in the ice core time series presented in SCD Figure 3a, when BC drops entirely (annual) or by ~50% (5 yr) from 1880 to 1890. Likewise, no explanation is given as to the steep drop from ~1920 and then the enormous climb beginning in 1930, coincident with the global depression.

The low concentrations in the 1890s may represent an artefact of the kind we summarized in the section "Ice-core drilling site": "*On inter-annual timescales, the summer-biased and irregular snow deposition contributes to the observed variability of the proxy records, with occasionally preserved winter snowfall typically having low impurity concentrations*". This is a site-specific characteristic resulting most likely from increased preservation of winter snow in the 1890s with strongly depleted d$^{18}$O values and overall low impurity content that has been previously discussed for other ice cores from Colle Gnifetti (Wagenbach et al., 2012). On the basis of conservative aerosol proxies (such experiencing no major trends; e.g. dust, sea-salt) and the ratio of stable isotopes in water we can identify that this glacio-chemical anomaly is unique in the context of the record discussed here (starting in 1741 AD).

Low BC concentrations are centred on 1930 before BC starts to increase again in 1933. Similar trends have been previously described for lead and were explained with the Great Depression starting in 1929 and lasting into the 1930s (Gabrieli and Barbante, 2014).

We added the Wagenbach et al. reference describing that the 1890 AD period experienced anomalous high winter snow preservation at Colle Gnifetti.

The timing of these excursions is markedly out of phase with well understood historical excursions in continental European productivity. Regarding the nonquantitation between the core and regional emissions, the interannual variation in the laser-incandescence (LI) data is quite large. SCD contends that the linear interpolation between years in the Bond et al emission estimates is not credible. Here, we show European emissions data for 2000-2015 (Figure 3), which span the largest economic variation in more than half a century. Note that PM2.5 emissions still had an average interannual variation of 3 percent. This suggests that the noise in the ice core BC concentrations is in transport variation year to year, not to first order emissions.

It is obvious that any measurement of a variable at a single given location experiences more variability than emissions integrated over all of Europe. There is some additional noise in the ice-core data due to inter-annual variability of transport, but the long-term trend is not affected as illustrated by the agreement between emission estimates and ice core data for sulphate (Engardt et al., 2017).

The burden needs to be put back to the authors to defend the "emergence" statistic. Indeed, the more noise, the later the lag in detection of any increase in LI-sensitive carbon. These thoughts are detailed in the four points below:

We have performed ToE analyses explicitly after removing forest fire related BC peaks during the pre-industrial in order to reduce the "noise" from non-industrial sources. Performing BC on the original dataset identifies 1895 or 1906 depending on the choice of the method as "time-of-emergence" (Supplementary Fig. S7). We also repeated ToE analyses for the four ice cores from Greenland and identified "time-of-emergences" in the range of 1864 to 1878 AD (1886-1892 AD using the Bayesian changepoint method). No matter which proxy and parameters we used to infer industrial BC emissions we found no evidence that "*BC emissions increased dramatically after 1850*" (Painter et al., 2013) which is the key assumption in the PP13 hypothesis.
We now report the results of ToE analyses performed for Greenland ice cores and added a new Supplementary Figure S7.

1. The use of summer-only data assumes that summer aerosols at the Col represent the operative aerosol deposition impacting the whole glacier - the strong persistent inversion in the Alps in summer disconnects the atmosphere at the Col from the atmosphere over the transport and ablation zones of the valley glaciers (Figure 4), thus the Col record is not a quantitative analog of valley conditions, and there is no evidence presented that winter and spring deposition can be inferred from the summer data.

This photograph is neither representing typical conditions prevailing throughout summer nor is this claim above substantiated by any robust data. Only Colle Gnifetti has a summer bias, whereas the Fiescherhorn ice core contains snow from all seasons. Yet, both records show overall similar long-term trends with BC concentrations at pre-industrial levels until 1875 AD. As shown in the Figure below, every summer, BC and other aerosols from the lowlands get frequently deposited at sites as high as Colle Gnifetti (Lugauer et al., 1998). Unless one argued that the industrialization in the regions surrounding Colle Gnifetti

produced emissions only during winter-spring, we don't see a pathway in which valley glaciers should have experienced a different long-term history of BC deposition than the sites where we extracted the ice cores.

[Figure]

*Figure. Monthly median (symbol) and quartile values (whiskers) of the epiphaniometer signal monitoring aerosol particles at the Jungfraujoch and at the Colle Gnifetti (only medians) from April 1988 to September 1996. (Lugauer et al., 1998)*

2. The laser incandescence (LI) method is highly sensitive to carbonaceous chain aggregate particles and is usually calibrated with a pure hydrocarbon combustion soot generator. The measurement is less sensitive to "brown carbon" and mixed-component aerosols such as produced by burning low quality coal or inefficient coal combustion (Sun et al., 2017). It may be that the record reflects evolving combustion technology as much or more so than emission strength.

We used both "elemental carbon" and "black carbon" each determined with different instrumentation and also other elemental coal burning proxies (e.g. Bi), but none of the measurements indicate significant increases before 1870. We cannot disentangle the relative importance of evolving combustion technology versus emission strengths since both parameters are largely unconstrained during the 19th century.

3. The large inter-annual variation in the LI data is inconsistent with interpreting it as a surrogate for regional or continental scale emissions. Data on modern fine particle emissions from 2000 to 2015 (Figure 3) show an average inter-annual variation of about 3%, even though this period includes the enormous economic fluctuations driven by the crisis of 2008. This suggests a large variability in the LI data due to meteorology. Hence, these data suggest noise is in transport variation year to year, not emissions.

We do not recommend to interpret interannual BC variability from a single ice-core site as a quantitative reconstruction of large-scale emissions but notice that the absence of any significant increases of BC between 1850 AD and 1875 AD in virtual all ice cores in Greenland and the Alps are inconsistent with 1) the idea of linearly rising BC emissions in North America and Europe between 1850 and 1900; and 2) with the hypothesis of BC constituting the main driver of Alpine glacier recession on the assumption that "*black carbon concentrations increased abruptly in the mid-19th century*" (Painter et al., 2013)

4. The emergence test statistic is strongly influenced by the noise in the time series data - with detection of a trend delayed as noise increases. Since the noise in these data contains a large meteorological component, the inferred timing of the beginning of industrial impact in these data has a large and undiscussed time lag.

While there are limitations in any methodology to estimate break-points in time-series depending on the noise and also other factors we have tried to estimate a lower (i.e. older) bound by "cleaning" the BC data for occasional forest fire peaks (to reduce the noise) and by including conservative age error estimates. Mean BC concentrations for the time period 1850-1875 AD are less than 20% elevated compared to the long-term preindustrial mean (1790-1850 AD) at Colle Gnifetti, Fiescherhorn and four different Greenland ice core sites, respectively.

The time of emergence results that we present are based on a distribution of results that assess the sensitivity of the ToE to methodological choices in the reference window length. We also further verify that similar late 19th Century changepoints are produced for the GC and Greenland ice core data using a Bayesian changepoint method (Ruggieri 2013).

Further, even assuming that deposition processes affect the BC preserved in the ice cores resulting in a "noisier" signal than for European-wide emissions (which we would expect), this "noise" would equally apply to the albedo effect of BC acting upon an individual glacier. The time when the concentration of BC in the ice core rose above the level of pre-industrial natural variability is thus a relevant measure for assessing when an unusual forcing by BC on the glacier could have become a plausible driver of glacier length changes.

Note also that in SCD Figure 5, the climb in BC in Greenland unambiguously occurs around 1850-1860.

In Figure 5, the unambiguous climb above background variability occurs only after 1870 AD. The period 1850-1860 was characterized by numerous forest fires (e.g. 1854, 1863, 1868, 1871) as can be proven for D4 using vanillic acid (VA) as a distinctive forest fire tracer (McConnell et al., 2007). For clarification, we repeated ToE analyses for all four individual ice cores: D4: 1878 [1865-1885; 5-95%], Summit2010: 1873 [1865-1878], NEEM: 1872 [1830-1882], TUNU: 1864 [1857-1869]. McConnell et al. (2007) give 1888 AD as their best estimate for industrial BC emergence at D4 which closely matches the dates determined

using the Bayesian changepoint method (1883-1893 AD, 5-95% range of all four ice cores).
We added a new figure Supplementary Fig. S7 showing the ToE analysis results for the D4 ice core – the only ice-core with annual dating accuracy, and a clear-cut proxy (vanillic acid) for forest-fire activity, respectively (McConnell et al., 2007) allowing discrimination of forest-fore BC.

(3) Radiative Transfer
SCD do not address the magnitude of radiative forcing by BC that was present in the Alps at the time. For the claim of no role for BC in the retreat of European glaciers from the LIA to be valid, the radiative forcings in Painter et al (2013) estimated from the ice core BC would have to be overestimated by more than an order of magnitude. We think this is highly unlikely given the known increase in aerosol concentrations with decreasing altitude, as substantiated by SCD co-author Schwikowski (2004), among many others. From an energy balance perspective, we do not understand how SCD would explain away the 20-40 W m-2 seasonally-averaged radiative forcing by BC for April-June in the ablation zones? As described in Painter et al (2013), the melt magnitude associated with the 1880 radiative forcing of 10-20 W m-2 would have been 240-480 kg m-2 (0.6-1.2 m w.e.) and with the 1900 radiative forcing of 19-38 W m-2 would have been 450-890 kg m-2 (1.1-2.2 m w.e.). Equivalent changes in temperature to produce such radiative forcings in context of mass balance would have reached 3-4 K.

Basis of these calculations were 1) not reproducible BC concentration estimates from an earlier study (Thevenon et al., 2009) which used an analytical setup that has not been used by any other research group since, and 2) largely unconstrained estimates of altitudinal gradients between valley glaciers and ice-core glaciers in the order of a factor of 10-20.  If these previous estimates (e.g. 10-20 W/m$^2$; 0.6-1.2 m weq) were indeed realistic representations of the actual radiative forcing in 1880 AD, the radiative forcing from 1910-1920 AD (when industrial BC content was three times that of 1875-1885 AD) and the resulting glacier melt rates would have been extreme, but overall glaciers advanced during this time period.

SCD state that our study suffers from not including the radiative forcing by dust in snow (p3, lines 32-35+p4, 1-2). To a degree they are correct because we did not include the description of our coupled dust+BC radiative forcings. However, the BC radiative forcings that we report were in fact (BC+Dust)-(Dust Only). Moreover, we concluded from extensive analyses of these cores that dust deposition saw no trends during the period recorded in the ice cores. Likewise, SCD also found that mineral dust had no trends (p8, 30-32), "In agreement with other dust records from Colle Gnifetti (Bohleber et al., 2018; Wagenbach and Geis, 1989), we observe no enhanced mean (or frequency) of mineral dust deposition throughout the 19th century (Supplementary Figs. S1, S4). "Either way, we are puzzled that such would be highlighted as a 'suffering'.

We agree that dust has no long-term trends in the 19[th] century but this had not been explicitly addressed in the PP13 paper. In hindsight, we consider this a minor issue in the PP13 paper.

Summary
Again, we are enthused that our work has stimulated thought and testing of our hypothesis. However, based on the above comments, it appears to be clear that SCD does not refute the hypothesis. Ultimately, the core of the SCD paper is really to reconcile an issue with ice cores and emission scenarios, neither of which presently can be considered quantitatively robust in characterizing regional atmospheric conditions at the elevations of Alpine glacier ablation zones (today back to the high stand of the LIA). Instead, the ice cores can be considered as suggestive of BC deposition timing and magnitude but not as an absolute quantification. Likewise, to a lesser degree the paper also encounters the discrepancy between the temperature observations, HISTALP reconstructions, and the tree-ring derived temperature reconstruction of Büntgen et al (2012). Let's work as a community to resolve these particular discrepancies.

The last sentence excluded, we respectfully disagree with this interpretation. The basis of the BC forcing hypothesis was that increased glacier retreat rates were coinciding with abrupt increases of industrial BC since 1850 based on model experiments using ice-core records, whereas temperatures appeared to have shown a declining trend throughout the 19[th] century. Using new measurements, we demonstrated that glacier melt accelerated before industrial BC emissions started to rise and highlighted the role of natural glacier variations resulting from global-scale radiative forcing from volcanic eruptions between 1600 and 1840 AD. We agree that detection and attribution of individual drivers of climate and glacier variability requires a community effort but also more and better observations and proxies. With the data at hand we, however, cannot identify a leading role in industrial BC in ending the "Little Ice Age". Previous *'Little Ice Age'*-type events (e.g., the *Late Antique Little Ice Age*) also ended, eventually (Büntgen et al., 2016).

References:
Bond, T. C., Bhardwaj, E., Dong, R., Jogani, R., Jung, S., Roden, C., . . . Trautmann, N. M. (2007). Historical emissions of black and organic carbon aerosols from energy-related combustion, 1850-2000. Global Biogeochemical Cycles, 21(2), GB2018. doi:10.1029/2006GB002840
Bond, T. C., Doherty, S. J., Fahey, D. W., Forster, P. M., Berntsen, T., DeAngelo, B. J., . . . Zender, C. S. (2013). Bounding the role of black carbon in the climate system: A scientific assessment. J. Geophys. Res. doi:10.1002/jgrd.50171
Farinotti, D. (2013). On the effect of short-term climate variability on mountain glaciers: insights from a case study. J. Glaciol., 59(217), 992-1006. doi:10.3189/2013JoG13J080
Gabbi, J., Huss, M., Bauder, A., Cao, F., & Schwikowski, M. (2015). The impact of Saharan dust and black carbon on albedo and long-term glacier mass balance. The Cryosphere, 9, 1133-1175. doi:10.5194/tcd-9-1133-2015

Grove, J. M. (2004). Little ice ages: ancient and modern (Vol. 1 and 2). London: Routledge.

Huybrechts, P., Nooze, P. d., & Decleir, H. (1989). Numerical modeling of Glacier d'Argentiere and its historical front variations. In J. Oerlemans (Ed.), Glacier Fluctuations and Climatic Change (pp. 373-389): Kluwer Academic Publishers.

Kerschner, H. (1997). Statistical modelling of equilibrium-line altitudes of Hintereisferner, central Alps, Austria, 1859-present. Ann. Glaciol., 24, 111-115.

Lavanchy, V. M. H., Gaggeler, H. W., Schotterer, U., Schwikowski, M., & Baltensperger, U. (1999). Historical record of carbonaceous particle concentrations from a European high-alpine glacier (Colle Gnifetti, Switzerland). J. Geophys. Res., 104(D17), 21,227-221,236. doi:10.1029/1999JD900408

Painter, T. H., Flanner, M., Marzeion, B., Kaser, G., VanCuren, R., & Abdalati, W. (2013). End of the Little Ice Age in the Alps forced by black carbon. Proc. Nat. Acad.Sci. USA. doi:10.1073/pnas.1302570110

Schwikowski, M. (2004). Reconstruction of European air pollution from alpine ice cores. In L. D. Cecil (Ed.), Earth Paleoenvironments: Records Preserved in Mid- and Low-Latitude Glaciers (pp. 95-119). Netherlands: Kluwer Academic.

Sun, J., Zhi, G., Hitzenberger, R., Chen, Y., Tian, C., Zhang, Y., . . . Mo, Y. (2017). Emission factors and light absorption properties of brown carbon from household coal combustion in China. Atmospheric Chemistry and Physics, 17, 4769-4780. doi:10.5194/acp-17-4769-2017

**Reviewer #2 (RC2):**

This paper evaluates the hypothesis that black carbon deposition could play a dominant role in European glacier dynamics. The paper presents black carbon concentrations found in ice cores at Colle Gnifetti along with other tracers of different kinds of combustion. Authors examine the role of mineral dust in possible forcing; since it absorbs light, mineral dust can be a possible confounding factor. They also compare their measurements with those of other ice cores. Finally, they compare the trends of black carbon deposition with those of black carbon emission and point out discrepancies between measured tracers and bottom-up inventories. The work is supported by a careful treatment of timing and uncertainties to interpret the ice core measurements.

We kindly acknowledge this positive and constructive evaluation. We carefully considered the improvements suggested below and we have revised the manuscript accordingly.

Overall, this part of the paper is a quite thorough and welcome contribution to the discussion of black carbon (and other species) emissions and influence during the industrial era. I commend the authors on their careful work. The next part of the work– and the origin of the paper's title– compares the timing of glacial retreat with the timing of black carbon increase. These glaciers are frequently observed, making them good candidates for such an analysis. Authors identify retreat and advance periods and compare them with time-of-emergence of black carbon above pre-industrial periods, finding no relationship. Finally, on page 11, authors posit that volcanic forcing, and not change in albedo caused by black carbon deposition, is the cause of glacial retreat. The glaciers analyzed (four in a "glacier stack") begin to retreat before the increase in black carbon emissions, so it is unlikely that black carbon, alone, caused the current retreat. Despite this good point, this part of the work appears less supported by the evidence presented. An important question is on what time and spatial scales one respects a response between a forcing and a regional response. Attribution of climate response typically involves some kind of large-scale pattern matching, considering more factors than given here. One doesn't expect an increase in black carbon emission to correlate neatly with the glacier retreat– although certainly the fact that glaciers retreated first indicates that other causes are at work. If there were such neat correlations, we should have much less trouble identifying the causes of climate change, overall. So the following questions would have to be answered in order to confidently state the "No role for black carbon" as in the title: What other factors could contribute to glacial retreat; How much do they vary and on what temporal and spatial scales (i.e. what noise could confound the signal and must be averaged out); and then how much black carbon does contribute and whether it has a significant effect. The authors also considered volcanic forcing, which they suggest to be much more relevant than black carbon forcing, yet they did not provide any quantification of or data behind the volcanic forcing, but only some discussion. That quantification would be needed in order to make the statements in this paper with confidence.

We fully agree with the reviewer. We did not intend to perform a classical *Detection-and-Attribution* study delineating the changing contributions of natural and anthropogenic climate forcers on European glacier dynamics through time. With growing emissions of greenhouse gases and anthropogenic aerosols starting in the 19th century a monocausal relationship between glacier dynamics and any single forcing is inherently difficult to detect, let alone to quantify. However, there is also growing scientific evidence of the strong influence of natural forcing (e.g., from volcanic eruptions) on regional climate (temperature, precipitation) a fact that had been largely ignored by Painter et al., (2013) who argued in favour of an exclusive role of anthropogenic snow-albedo forcing from industrial soot.

To better reflect that we did not quantify relative contributions of forcing to glacier variability we changed the title to "*No **leading** role for industrial black carbon in forcing 19th century glacier retreat in the Alps*"

The four glaciers we used in our manuscript have mean "reaction times" of the glacier tongue to a climate perturbation of on average less than 10 years, whereas the "response time", the time required for a glacier to adjust from one "steady-state" to another, following a change in the mass balance may be two to three times longer (Nussbaumer & Zumbühl. 2012). However, the reaction to atmospheric conditions at the glacier front can also be more immediate in some situations, e.g. after runs of cool summers as becomes evident by maximum glacier advances in 1602, 1644, 1821 following "volcanic winters" with 5 years delay or less. To err on the side of caution, we have, in the interpretation of our "time-of-emergence" assessment, assumed that glacier fronts would have reacted immediately to increased rates of BC deposition.

In the revised manuscript, we put volcanic aerosol forcing from 1600-1840 AD in a long-term perspective. We quantified that Northern hemisphere (30-90°N) stratospheric aerosol optical depth (SAOD) was during this time 40% higher than the mean of the Common Era. Strong aerosol forcing culminated in a cluster of large eruptions during the early 19th century that was followed by Alpine glacier advances until the mid-19th century (new Fig. 9). New paleo-reanalyses covering the past 400 years confirm the large-magnitude, large-scale cooling impact of explosive eruptions Franke et al., (2017).

[Figure]

*Figure. Northern hemisphere temperature evolution over the past centuries (Franke et al., 2017)*

**References:**

Andres, R. J., Fielding, D. J., Marland, G., Boden, T. A., Kumar, N., and Kearney, A. T.: Carbon dioxide emissions from fossil-fuel use, 1751-1950, Tellus B, 51, 759-765, 1999.

Bohleber, P., Erhardt, T., Spaulding, N., Hoffmann, H., Fischer, H., and Mayewski, P.: Temperature and mineral dust variability recorded in two low-accumulation Alpine ice cores over the last millennium, Clim Past, 14, 21-37, 2018.

Böhm, R., Jones, P. D., Hiebl, J., Frank, D., Brunetti, M., and Maugeri, M.: The early instrumental warm-bias: a solution for long central European temperature series 1760-2007, Climatic Change, 101, 41-67, 2010.

Büntgen, U., Tegel, W., Nicolussi, K., McCormick, M., Frank, D., Trouet, V., Kaplan, J. O., Herzig, F., Heussner, K. U., Wanner, H., Luterbacher, J., and Esper, J.: 2500 Years of European Climate Variability and Human Susceptibility, Science, 331, 578-582, 2011.

Büntgen, U.; V. S. Myglan, F. Charpentier Ljungqvist, M. McCormick, N. Di Cosmo, M. Sigl, J. Jungclaus, S. Wagner, P. J. Krusic, J. Esper, J. O. Kaplan, M. A. C. de Vaan, J. Luterbacher, L. Wacker, W. Tegel, A. V. Kirdyanov. Cooling and societal change during the Late Antique Little Ice Age (536 to around 660 CE), Nature Geoscience 9, 231–236 (2016)

Engardt, M., Simpson, D., Schwikowski, M., and Granat, L.: Deposition of sulphur and nitrogen in Europe 1900-2050. Model calculations and comparison to historical observations, Tellus B, 69, 2017.

Frank, D., Büntgen, U., Böhm, R., Maugeri, M., and Esper, J.: Warmer early instrumental measurements versus colder reconstructed temperatures: shooting at a moving target, Quaternary Sci Rev, 26, 3298-3310, 2007.

Franke, J., Brönnimann, S., Bhend J. & Brugnara Y.: A monthly global paleo-reanalysis of the atmosphere from 1600 to 2005 for studying past climatic variations, Scientific Data, 4:170076, doi:10.1038/sdata.2017.76, 2017.

Gabrieli, J. and Barbante, C.: The Alps in the age of the Anthropocene: the impact of human activities on the cryosphere recorded in the Colle Gnifetti glacier, Rend Lincei-Sci Fis, 25, 71-83, 2014.

Holzhauser, H., Magny, M., and Zumbuhl, H. J.: Glacier and lake-level variations in west-central Europe over the last 3500 years, Holocene, 15, 789-801, 2005.

Hong, S. M., Candelone, J. P., Patterson, C. C., and Boutron, C. F.: Greenland Ice Evidence of Hemispheric Lead Pollution 2-Millennia Ago by Greek and Roman Civilizations, Science, 265, 1841-1843, 1994.

Lugauer, M., Baltensperger, U., Furger, M., Gaggeler, H. W., Jost, D. T., Schwikowski, M., and Wanner, H.: Aerosol transport to the high Alpine sites

Jungfraujoch (3454 m asl) and Colle Gnifetti (4452 m asl), Tellus B, 50, 76-92, 1998.

Luterbacher, J., et al.: European summer temperatures since Roman times, Environ Res Lett, 11, 2016.

Lüthi, M. P.: Little Ice Age climate reconstruction from ensemble reanalysis of Alpine glacier fluctuations, Cryosphere, 8, 639-650, 2014.

McConnell, J. R., Wilson, A. I., Stohl, A., Arienzo, M. M., Chellman, N. J., Eckhardt, S., Thompson, E. M., Pollard, A. M., and Steffensen, J. P.: Lead pollution recorded in Greenland ice indicates European emissions tracked plagues, wars, and imperial expansion during antiquity, Proceedings of the National Academy of Sciences, doi: 10.1073/pnas.1721818115, 2018. 2018.

McConnell, J. R., Edwards, R., Kok, G. L., Flanner, M. G., Zender, C. S., Saltzman, E. S., Banta, J. R., Pasteris, D. R., Carter, M. M., and Kahl, J. D. W.: 20th-century industrial black carbon emissions altered arctic climate forcing, Science, 317, 1381-1384, 2007.

Miller, G. H., Geirsdottir, A., Zhong, Y. F., Larsen, D. J., Otto-Bliesner, B. L., Holland, M. M., Bailey, D. A., Refsnider, K. A., Lehman, S. J., Southon, J. R., Anderson, C., Bjornsson, H., and Thordarson, T.: Abrupt onset of the Little Ice Age triggered by volcanism and sustained by sea-ice/ocean feedbacks, Geophys Res Lett, 39, 2012.

Nussbaumer, S. U. and Zumbühl, H. J.: The Little Ice Age history of the Glacier des Bossons (Mont Blanc massif, France): a new high-resolution glacier length curve based on historical documents, Climatic Change, 111, 301-334, 2012.

Nyeki, S., Kalberer, M., Colbeck, I., De Wekker, S., Furger, M., Gäggeler, H. W., Kossmann, M., Lugauer, M., Steyn, D., Weingartner, E., Wirth, M., and Baltensperger, U.: Convective boundary layer evolution to 4 km asl over high-alpine terrain: Airborne lidar observations in the Alps, Geophys Res Lett, 27, 689-692, 2000.

Robock, A. and Mao, J. P.: The Volcanic Signal in Surface-Temperature Observations, J Climate, 8, 1086-1103, 1995.

Schurer, A. P., Tett, S. F. B., and Hegerl, G. C.: Small influence of solar variability on climate over the past millennium, Nat Geosci, 7, 104-108, 2014.

Schwikowski, M. (2004). Reconstruction of European air pollution from alpine ice cores. In L. D. Cecil (Ed.), Earth Paleoenvironments: Records Preserved in Mid- and Low-Latitude Glaciers (pp. 95-119). Netherlands: Kluwer Academic.

Solomina, O. N., Bradley, R. S., Jomelli, V., Geirsdottir, A., Kaufman, D. S., Koch, J., Mckay, N. P., Masiokas, M., Miller, G., Nesje, A., Nicolussi, K., Owen, L. A., Putnam, A. E., Wanner, H., Wiles, G., and Yang, B.: Glacier fluctuations during the past 2000 years, Quaternary Sci Rev, 149, 61-90, 2016.

Thevenon, F., Anselmetti, F.S. Bernasconi S.M. and Schwikowski M., Mineral dust and elemental black carbon records from an Alpine ice core (Colle Gnifetti glacier) over the last millennium. Journal of Geophysical Research 114, D17102 (2009).

Toohey, M. and Sigl, M.: Volcanic stratospheric sulfur injections and aerosol optical depth from 500 BCE to 1900 CE, Earth System Science Data, 9, 809-831, 2017.

Usoskin, I. G.: A history of solar activity over millennia, Living Rev Sol Phys, 14, 2017.

Wagenbach, D., Bohleber, P., and Preunkert, S.: Cold, Alpine Ice Bodies Revisited: What May We Learn from Their Impurity and Isotope Content?, Geogr Ann A, 94a, 245-263, 2012.

Zumbühl, H. J., Steiner, D., and Nussbaumer, S. U.: 19th century glacier representations and fluctuations in the central and western European Alps: An interdisciplinary approach, Global Planet Change, 60, 42-57, 2008.

---

## Editor Decision (ED1)

Dear Dr. Sigl,

Thank you for your consideration of the 3[rd] reviewer's comments and for your revisions. I read through your replies and revisions carefully. I think that you mostly (but not entirely) have addressed the reviewers' comments. Your additions to the manuscript were a bit frustrating to read as they contained many overly long, run-on, grammatically incorrect sentences. I have noted this specifically below. I've also suggested some wording changes that I think will better respond to the reviewers' concerns and make the manuscript more clear. Please submit your revisions as tracked changes to me after carefully reading through and editing your writing. Page and line numbers refer to the tracked changes version of your revisions.

Regards,

Becky Alexander

Title:

The title is too long. It would be better to simply say "19[th] century retreat in the Alps preceded the emergence of industrial black carbon deposition on high-alpine glaciers" as that is what your paper clearly shows.

Abstract:

Page 1 Line 26: What does "well-replicated" mean? This is vague. Did you do replicate analysis of samples or is based on your stacked data set? I think it's the latter, but it would be better to choose more clear wording.

Page 1 lines 32 to end of abstract: I suggest ending the abstract as follows beginning on line 32: "…than 80% of their total 19[th] century length reduction, casting doubt on a leading role for anthropogenic BC emissions in terminating the Little Ice Age. Attribution of glacial retreat requires expansion of spatial network and sampling density of high alpine ice cores to balance potential biasing effects arising from transport, deposition and snow conservation in individual ice core records." I don't think the reviewers will be happy with the sentence beginning on line 32 ("Industrial BC emissions…") as you have not done any forcing calculations as the sentence may imply. The sentence beginning on line 33 ("BC records…") comes out of nowhere and doesn't make sense here.

Introduction:

Page 2 Line 11-12: Change to "…because it absorbs solar radiation even at very low concentrations…"

Page 3 lines 1-4: This is an example of a long run-on sentence that is difficult to make sense of. It needs to be split into two sentences. Perhaps "Together with mineral dust and other absorbing organic aerosols, BC deposited on snow and ice can lead to increased melt rates and changes in melt onset due to reductions in surface albedo. These effects are further enhanced by subsequent snow albedo feedbacks such as an increase in the water content and surface accumulation of impurities."

Page 3 line 5: Should read "…best estimate for industrial era global forcing of BC is…"

Page 4: This new section is difficult to read. The first sentence is long and grammatically incorrect. I cannot tell exactly what you are trying to say. I'll leave it up to you and your co-authors to split this up into two or more sentences and check it for grammar and correctness. The next section beginning with "The snow-albedo feedback" on line 18 also contains long, run-on sentences and is very disorganized making it difficult to read. You are casting doubt on the Painter et al study for three reasons: 1. A low resolution data set, 2. Measurements were performed using questionable analytical methods and 3. Large dating uncertainties. Perhaps you could start it like this: "The snow-albedo feedback hypothesis formulated by Painter et al was a first effort to attempt this but was limited by the available BC data at that time[here provide references for the data that was used in this study]. The available data were of relatively low time-resolution [reference specific data set(s)] and the dating was uncertain [reference specific data set(s)]." The phrase "compiled from very first applications using different analytical methods" doesn't really say anything. Just because they are the "first" measurements or "different" (different from what?) doesn't mean they aren't any good. You need to be specific on what you think is wrong with the analytical technique and provide appropriate references to support this. "Some of these methods" – which methods? Both Jenk et al and Thevenon et al or just one of them? At this point the reader doesn't know how many data sets you are referring to and if they all suffer from the same issues. How do you know that a method(s) did not deliver reproducible results? Is there a reference for this? Personal communication? You need to support such a claim. Perhaps you should follow this "The snow-albedo feedback hypothesis formulated by Painter et al was a first effort to attempt this but was limited by the available BC data at that time[here provide references for the data that was used in this study]. The available data were of relatively low time-resolution [reference specific data set(s)] and the dating was uncertain [reference specific data set(s)]." with "Here, we set out to re-evaluate the timing of industrial BC deposition in ice cores from the alps by using a new, more accurately dated record of light absorbing aerosols at much higher time resolution (sub-annual). In addition, we measure distinctive tracers of anthropogenic pollution (list them here) and compare all records with the most highly resolved history of glacier length changes of four glaciers in the Western Alps currently available." This removes some of the details (which are unclear for reasons stated above), and I suggest moving this more detailed discussion into section 3.2 I also suggest removing the last sentence of this section beginning with "We determine…"

Section 3.2:

Page 10 line 20: What do you mean by "non-equidistant samples"? "not-validated method" is not supported – see also my previous comment from the introduction.

Page 10 line 23: remove comma after "14 years"

Page 10 line 26: Should read "reach comparable levels during the peak"

Page 10 line 27: Replace "assume" with "speculate"

Discussion:

Page 13 lines 8 – 9: Should read "82% [52%] of the glacier length reductions had already occurred at the best [earliest] estimated time of emergence of industrial BC deposition."  I removed the end of this sentence because you can't say it played no role later on.

Page 13 line 12: "1900 AD AND also by"

Page 13 line 14: remove comma after 1875 AD

Page 13 line 19: insert comma after Sun et al reference.

Page 13 lines 20-24: Should read: "However, these species are measured by the method applied for the EC record from FH02 which shows very good agreement with the rBC record from Colle Gnifetti (Section 3.2).  This suggest that factors other than changes in surface snow albedo, such as temperature and seasonal precipitation distribution (Steiner, Zumbuhl), may have dominated mass balance and glacier length variability of these European glaciers until at least 1875 AD."

Page 13 line 26: remove comma after variability

Page 14 line 10: change "can thus be understood as" to "may be"

Page 14 line 14: remove "to some extent".

Page 14 line 19-21: Should read "…will require reconciliation of early instrumental and proxy climate data [references] and the use of models to decompose…"

Page 14 lines 25-29: Remove the last 3 sentences.  This belongs in the conclusions.

Conclusions:

Page 15 line 28: replace "enhancing" with "reductions in"

Page 15 line 32-33: Should read "species co-analyzed enabled BC source attribution from industrial and biomass burning emissions."

Page 16 line 7: Should read "We hypothesize that glacier length changes throughout the past 2,000 years have been…"

Page 16 line 15: Remove last sentence of this paragraph.

Page 16 line 16: Replace "computer" with "model"

Page 16 line 18: Should read "the past few decades"

Page 16 line 20: Should read "model evaluation" (remove the "s")

Page 16 lines 21-22: Should read "Here we present the first continuous…"

Page 16 lines 24-27: Should read "Aerosol deposition at any single site also depends on factors such as atmospheric transport efficiency and the spatial distribution and conservation of snowfall. Incorporating more BC records from multiple sites into a stacked composite is expected to enhance the

signal from the atmospheric burden over the noise caused by spatial variations in atmospheric transport and snow accumulation."

---

## Author Response (AR2)

**Reply to Anonymous Referee #3**

We want to thank again all three reviewers for their detailed evaluation of our work, and for their constructive commentary that has helped us improve the manuscript. The comments of Reviewer 3 are printed in *green italic* font, with our responses in black. Figures that are part of this response (but not included in the revised manuscript) are given at the end of this document, and have an "R" in their label (for example Figure R1).

*--------------- Referee #3 (Remarks to the Author): ------------------------------------------------*

*General Comments*

*The authors present a careful analysis of the timing of refractory black carbon (rBC) deposition in the Alps of western Europe based on duplicate ice cores. The record of rBC from this core in the late 19th Century is significantly different from that obtained in an earlier ice core (Thevenon et al., 2009) and indicates the deposition of rBC did not significantly increase beyond the 'natural' pre-industrial variability until at earliest 1868. The authors use this observation to attempt to refute the hypothesis that industrial emissions of black carbon may have been responsible for forcing the synchronous retreat of glaciers at the end of the little Ice Age in the European Alps (Painter et al. 2013). The authors compare the new record to similar records from ice cores elsewhere in the Alps and in Greenland. They also compare the records of rBC deposition with bottom-up emission estimates. While the authors clearly demonstrate that statistically significant increases in rBC deposition occurred after the generally accepted timing of the onset of glacier retreat in the Alps, they do not present conclusive evidence that rBC did not play a role in this retreat. Specifically, they have not used methods that can clearly demonstrate the role (or lack of role) of rBC in forcing glacier fluctuations during this period. To do this would require modelling the often-complex relationships between glacier fluctuations and meteorological forcing, considering glacier feedbacks (e.g. albedo feedback) and the effect of rBC on albedo, to demonstrate the mechanisms responsible for retreat in the second half of the 20th Century (e.g. Zekollari 2017, or Goosse et al., 2018). The authors seem to be looking for a mono-causal relationship between glacier fluctuations and rBC forcing over the past centuries, rather than accounting for the dynamic relationships between various meteorological forcing and glacier feedbacks and demonstrating the effect of each on forcing glacier fluctuations. This point was raised in the initial review by Reviewer 2, and the authors have not made significant revisions in response. The addition of longer records of glacier fluctuations and volcanic aerosols provided (in new Fig. 9), do not provide any quantification of the influence of different factors on glacier length fluctuations, as requested by the reviewer. There are no significant changes to the discussion or conclusions (other than discussion of longer aerosol and glacier records), and the title reports a result that has not been thoroughly tested in the paper. Either much greater revisions to the text are needed or additional analyses are needed to support the arguments.*

We did not intend to convey a message that any external forcing (be it industrial soot or volcanic eruptions) was in a mono-causal relationship linked to past glacier variations. Our strong title and the manuscripts original conclusion were to an extent driven by the very definite title "*End of the Little Ice Age in the Alps forced by industrial black carbon*" of the

original Painter et al. publication. In the light of our new study, this statement now lacks strong empiric support. While models are often used in detection and attribution studies, the validity of such studies depends also critically on the quality of the input data. In our study, we have contributed towards the goal of better understanding past glacier variability by developing the first reproducible ice-core record of rBC deposition from the Alps thereby also putting emphasis on the importance of accurate dating of such records. Our new results, significantly improved compared to the very first data available at the time of the Painter et al. study both due to the continuous development in analytical techniques in recent years and the improvement of accuracy in our ice-core chronology. Taken together, this allows us now to provide strong constraints on the timing of industrial soot deposition on alpine glaciers and thus testing of the hypothesis put out by Painter et al. The focus of this paper clearly lies on the timing of events and not on the modeling part, which we are happy to leave to the experts in this field in future studies. We are convinced that no one would argue about our main result and conclusion based on this focal point and knowing that a forcing factor happening after an apparent response cannot be its cause.

With the late 19th century experiencing the increase of anthropogenic GHGs, tropospheric aerosols such as sulfate and BC (all with different effects on climate) - and with the meteorological forcing (i.e., precipitation, air temperatures) not well constrained in high alpine, glaciated regions - it is inevitably difficult to discriminate - let alone quantify - the role of the various potential forcing agents and feedbacks on 19th century glacier variations.

We amended the revised manuscript to better reflect the limitations of the currently available methods and data sets in conclusively answering the open research question on the specific role of light absorbing substances on glacier length variability. We believe that additional analyses - e.g., radiative forcing calculations - are best addressed by the broader scientific community, as was also suggested in the SC by Painter and colleagues. Our new rBC reconstruction will provide a key constraint for such efforts.

We have made the following changes in response to this comment:
- We have added in the *Introduction* paragraph the importance of modeling efforts to attribute and quantify driving forces of glacier variations and the limitations of such efforts arising from the incompleteness/biases of previous observational constraints.
- We cite and discuss in the *Data and Methods* section previous work demonstrating the suitability of our ice-core site to faithfully record long-term trends of anthropogenic emissions from industrialized areas at low elevations.
- We have added in the *Results* section a more detailed comparison with the Colle Gnifetti EC record (previously used by Painter et al. 2013) focusing on the timing of long-term trends and the lack of demonstrated reproducibility for the EC results by Thevenon et al., (2009). We added one additional Supplementary Figure S7.
- In the *Discussion* paragraph we discuss limitations of currently available methods to quantify the full spectrum of light-absorbing impurities present in snow and highlight the need for integrative research to disentangle the driving forces for the end of the Little Ice Age in Europe.
- We changed the title and conclusion to better separate what our analyses actually proofs from what that might imply for the role of BC or any other potential forcing.

We decided to keep the new Figure 9 in the manuscript, because we believe there is great value to provide a long-term context of glacier variability under natural climate forcing.

*The authors certainly cast doubt on the hypothesis of Painter et al, 2013, but cannot take their interpretation as far as disproving it. It appears that rBC may not have played such a large role in the initial retreat of glaciers in the Alps before 1875 but it cannot be ruled out that they played a role or that they did not force the continued retreat in the 20th Century. The results presented here certainly highlight the need to reconcile differences in the air temperature estimates in the 19th Century from observational records, reconstructions and tree ring analysis, to enable confident attribution of observed glacier changes to different forcing.*

We fully agree with this evaluation. We would like to note, that in our manuscript we focus on the 19$^{th}$ century. A potential influence of BC on glacier retreat in the 20$^{th}$ century is out of the scope of this study and we do not doubt a potential influence during that time. We do clarify this in our new title and also in the reformulated sentences at the end of Section 4.1.

In our manuscript we wrote: "*The **glaciers' initial and more or less synchronous retreat from the maximum terminus positions starting at 1860** can thus be understood as a delayed rebound back to their positions they had before the radiative perturbed time period 1600-1840 AD (Fig. 9), and an additional decrease of snow albedo from the deposition of BC is considered not to be needed to explain these observations (Lüthi, 2014). The specific extent to which early anthropogenic warming (Abram et al., 2016), changes in atmospheric modes (Swingedouw et al., 2017) including the Atlantic Multidecadal Oscillation (AMO) (Huss et al., 2010), or to some extent snow-albedo feedbacks from **increasing light-absorbing aerosol deposition towards the end of the 19th century** may have contributed to the overall glacier length variability in the European Alps throughout the 19th century remains difficult to determine. *

We have replaced the last sentence in which we had previously rejected the hypothesis with a summary showing future research direction to tackle the research question: *"To confidently attribute and quantify the contribution of natural and anthropogenic forcing to observed glacier changes will require to reconcile early instrumental and proxy climate data (Böhm et al., 2010; Frank et al., 2007), and to use ensembles of simulations with a hierarchy of model complexities to decompose the relative contribution of volcanic eruptions, light-absorbing impurities such as BC or other compounds (brown carbon, mineral dust) and other potential natural or anthropogenic contributions (Goosse et al., 2018; Zekollari et al., 2014)."*

*The discussion of the validity of gradually increasing emissions during the 19th Century is a useful outcome of the manuscript. However, the authors need couch their discussion more careful in the context of the effect of meteorological variability on the relationship between emissions and the deposition records. This should include an evaluation of the influence of disconnection of the study site from the valley planetary boundary layer as suggested by Painter et al. 2013. To demonstrate that high frequency fluctuations (annual) in rBC can be interpreted as commensurate variations in emissions, an analysis of the relationship between contemporary emissions rates (which are more well constrained) and deposition rates at the field site is needed.*

BC concentrations at a single ice-core site are effected by meteorology, vertical transport snow conservation and emissions strengths. It is inevitable that records obtained from such locations show higher variability than emission estimates integrated on a country level. Most

studies that have related ice-core concentrations to trends in emissions suggested longer time periods (e.g., 5 or 10 years) must be averaged to compensate for seasonal and annual variability effects caused by vertical transport, snow redistribution and conservation. Long-term trends in sulfate (Schwikowski et al., 1999), nitrate (Engardt et al., 2017), lead (Schwikowski et al., 2004), polycyclic aromatic hydrocarbons (Gabrieli et al., 2010) and other species (Barbante et al., 2004) from the Colle Gnifetti ice-core **all closely follow estimated emission trends.** Similar results were reported from other high-alpine ice cores in the Alps (Fagerli et al., 2007; Preunkert and Legrand, 2013; Preunkert et al., 2003). We followed these recommendations (see Figures 3-6) in our manuscript and discuss primarily long-term trends (i.e., 5 or 10-year means, 11-year running means).

The studies summarized above, as well as results from long-term monitoring of aerosol transport to Colle Gnifetti and Jungfraujoch (Lugauer et al., 1998), all provide clear evidence that the ice-core site on **Colle Gnifetti is not disconnected throughout the summer from the valley planetary boundary layer** as was hypothesized by Painter et al., (2013). Polycyclic aromatic hydrocarbons emitted in the lowlands by coal burning are detected at Colle Gnifetti starting in 1895 AD (Gabrieli et al., 2010), as are micrometers-sized pollen grains, fly ash, wood fragments and other macroscopic material (Thevenon et al., 2009). There is no reason to believe that sub-micron sized rBC would not be transported and deposited on Colle Gnifetti, if significant emissions had taken place in the lowlands during the industrialization. Using the same analytical technique, rBC summer (defined here as all samples larger than the median) concentrations from the lower elevated Silvretta ice core (Switzerland, 2927 m asl; (Pavlova et al., 2015)) are only higher by a factor of 2.4 compared to the mean BC concentrations for Colle Gnifetti (4450 m asl) between 1964 and 1989 AD. While this is not a comprehensive analyses of the **altitudinal gradients** of aerosol deposition on snow, this is substantially less than the scaling factors of 10 to 20 that Painter et al., (2013) assumed to estimate equivalent concentrations at 2000 m (i.e., glacier front positions) from observed ice-core concentrations at 4000 m.

We further would like to note that in our manuscript we do compare our records of BC with the European emission estimates (which do have substantial uncertainties as well). The reasonable agreement indicates that our ice core records indeed allow reconstructing industrial BC emissions. Weather the deviations are due to a lack of the ice cores in their ability to record all variations or a lack in the accuracy of the emission estimates we cannot conclude but we discuss it rather extensively in Section 4.2. In any case, we would like to point out, that the initial hypothesis formulated by Painter et al. is based on the exact same study sites and cores but on the data available at that time which we have improved considerably since as presented here.

Anyhow, we think the discussion of the effect of meteorological variability on the relationship between emissions and the deposition records is important to address and we therefore have made the following changes in the manuscript:
- We have slightly extended the discussion of the relationship between emissions and deposition by including additional references that had previously demonstrated the ability to use ice-core concentrations (e.g. sulphate, nitrate, lead, PAHs, and others) obtained from Colle Gnifetti (and other alpine sites) as a first order surrogate of past emission strengths.
- We now discuss the idea of a potential disconnection between high and low elevation sites for nm-sized BC aerosols citing previous aerosols monitoring studies performed on Colle Gnifetti.

*The paper will certainly be an important addition to the literature but needs to be reframed and the discussion and conclusions changed to reflect the interpretations that can*

*reasonably be made from the data presented. In this light, the title is misleading and emphasises a more speculative implication of the research that has not been clearly demonstrated. I would suggest the title needs to be changed to align with its key finding e.g. "Increase in high-altitude industrial black carbon deposition occurred after initiation of 19th century glacier retreat in the Alps."*

We thank the reviewer for the suggestion. For the reasons discussed above we assume that long-term trends in BC deposition closely follow those of BC emissions. We also find the term "initiation" not appropriate since we calculated that 52-92% of the total 1850-1900 retreat had occurred prior to 1875 AD.

Anyhow, we agree with the comment that we do not do any quantification and therefore changed the title to: **"19ᵗʰ century glacier retreat in the Alps preceded the emergence of industrial black carbon deposition on high-alpine glaciers, casting doubt on a leading role for anthropogenic BC emissions in terminating the Little Ice Age"**

*Specific Comments (page-line):*
*p1 ln28 – "study reveals that in 1875 AD, the time when European rBC emission rates started to significantly increase" – but this has not been demonstrated – deposition at high altitudes increased at this time, but the question of emissions-deposition has not been addressed, so this sentence needs rewording.*
Changed to *"the time when rBC ice-core concentrations started to significantly increase"*

*p2 ln7 – "because it absorbs - in minute amounts - solar radiation" - this statement is ambiguous. Do you mean "even at very low concentrations, rBC absorbs solar radiation..." (as is, the sentence read that rBC on absorbs only small amounts of solar radiation).*
Changed as suggested

*p2 ln 29 – "involving snow albedo feedbacks that significantly enhance initial radiative forcing" – the use of 'radiative forcing' here is confusing as it is used earlier when referring to atmospheric forcing – suggest rewording to "that are significantly enhanced by snow albedo feedbacks"*
Changed as suggested

*p3 ln28 – this description of the BC records is very different from the description in Painter et al. 2013, who show a step change at Colle Gnifetti in 1850 – please explain the differences.*
Thevenon et al. (2009) used EC instead of rBC and describes in his paper a "large increase in 1875-1890". The "step change" described by Painter is composed of three elevated samples while in between the values drop to typical background concentrations. The largest EC peak on record (if samples are averaged to the same temporal resolution) is the increase dated by Thevenon et al. (2009) to 1875-1880 AD which also includes a major Saharan dust event. Please not that Thevenon et al. (2009) used a modeled timescale not forced through any stratigraphic age markers, thus lacking dating accuracy in the younger part (i.e., 1770-1940). During the 19ᵗʰ century, the mean offset relative to the annual-layer-counted chronology is 14 years (minimum 7, maximum 18 years). Large sample size requirements also did not permit to evaluate the reproducibility of their EC measurements. Biases in EC determination strongly depend on the method and are believed to be possible through for example charring of other organic compounds or through the presence of mineral dust (Lim et al., 2014).

We added a figure in supplement (Supplementary Fig. S7) that shows the EC record from Thevenon et al., (2009) clearly depicting the previous limitations regarding dating accuracy, lack of reproducibility and potential analytical biases.

*p3 ln32 – this discussion of the hypothesis of Painter et al. 2013 should be in the discussion section. Also, these limitations are not borne out by the data presented here (1) the new records show "the overall EC trend from Fiescherhorn ice core is closely reproduced by the new rBC record" – which would support the results of Painter et al. 2013, (2) – the paper does not demonstrate that the new records of glacier retreat differ significantly from those used by Painter et al. 2013, (3) – no data is presented to show that contributions from dust were important in the late 19th Century retreat.*

We decided to keep the discussion here, because we believe it is important to give context for the choice of the glacier records and the analytical instrumentation we used. However, we entirely reformulated this part, acknowledging the early work of Painter et al. and at the same time clarify the main focus of this paper and the significant benefit of the improved dataset we have at hand now to come to our conclusion.

We now write:

*The snow-albedo feedback hypothesis formulated by Painter et al. (2013) was a first effort to attempt this but was naturally limited by the BC data available at the time and thus compiled from very first applications using different analytical methods, performed in low temporal resolution and partly published on preliminary timescales only. Some of these methods turned out to be unable to deliver reproducible results and thus were not applied in any future investigation since. The obvious choice was to use glacier length curves with comparable low temporal resolution as the available BC records at the time did not allow resolving the precise timing of the BC increases. Clearly, their conclusions were also restricted by not considering contributions from other absorbing aerosols present in the snow (e.g., Saharan dust; brown carbon; macroscopic charcoal or pollen) which however are admittedly not well constrained (Brugger et al., 2018; Oerlemans et al., 2009; Skiles et al., 2012; Sun et al., 2017) and therefore might not have been incorporated into the radiative transfer model.*

*Here we set out to re-evaluate the timing of industrial BC deposition in ice cores from the Alps which is key information to assess the hypothesis of a strong role of industrial BC in forcing of the "End of the Little Ice Age in the Alps" (Painter et al., 2013). Years later, we are now in the fortunate position allowing us to do this by using a new, more accurately dated record of light absorbing aerosols in much higher resolution (i.e., BC and mineral dust), in combination with a suite of other distinctive tracers of anthropogenic pollution and in tandem with the most highly resolved history of glacier length changes of four glaciers in the Western Alps currently available (Nussbaumer and Zumbühl, 2012). We determine the time when industrial BC emerged from the preindustrial background relative to the observational history of terminus position of European glaciers and evaluate to which extent these results were consistent with the hypothesis put out for future investigations by Painter et al. (2013) of a strong ice-albedo feedback starting in the 1850s through increased ablation rates at the glaciers surface.*

Our new rBC record reproduces the Fiescherhorn (EC rises for the first time in 1875) but not the original Colle Gnifetti EC record (EC rises temporarily already in 1850) now discussed in more detail in the text (see also Supporting Fig. S7). The glacier observation density (in Painter et al. 2013 shown as dots) is small for Argentiere and Rhône glaciers during the 19[th] century. Missing data is essentially linearly interpolated over most of the late 19[th] century

obscuring the details of glacier length changes. In contrast, Bossons, Mer de Glace and Grindelwald glaciers have the highest observation densities over the time period of interest. Even without the new CG03 rBC record the mismatch in the timing between initial glacier retreat and EC rise at Fiescherhorn in 1875 AD becomes obvious using these glacier lengths curves (see Fig. R1).

[Figure]

*R1: High-resolution glacier length records and FH02 EC concentrations which started to rise in 1875 AD (green dashed line)*

We do not state that dust has played an important part in the glacier retreat, but mentioned that Painter has not previously assessed this possibility in their paper. We show the absence of increased dust deposition in Supplementary Fig. S2, new Fig. S7 and Fig. S10 (formerly S9). We rewrote the sentence to a more general statement that other potential light-absorbing impurities (including brown carbon) need to be taken into account for a more complete assessment.

*p4 ln2 - "Here we set out to rigorously re-evaluate the hypothesis of industrial BC forcing the "End of the Little Ice Age in the Alps" (Painter et al., 2013)". As discussed earlier, the manuscript does not use the appropriate methods to address this aim. I would suggest altering this aim and keeping the current analyses, rather than adding additional analyses that would address the current aim.*
We followed this suggestion (see comment above).

*p6 ln6 – this uncertainty needs to be addressed in the discussion as it is rather large compared to the short time between the time of emergence of rBC and the onset of glacier retreat.*
We have added the information in the caption of Figure 7. Our range of 52-92% for the glacier retreat that had occurred until ToE already takes the dating uncertainty into account.

*p8 ln8 – these correlation coefficients (0.35 and 0.63) are not high, especially for Pearson's correlation coefficients. Please reword.*
Changed as suggested. *"Due to common transport and deposition, co-variability of rBC with other species at intra-annual resolution is indicated by significant Pearson's correlation coefficients (p<0.001, one sided, N=696; $Na^+$, R=0.35; $NH_4^+$, R=0.63) for the period of the most recent 40 years."*

*p9 ln3 – here and in the discussion, the authors need to comment directly on how and why their record differs from the that in Thevenon et al. 2009. This is a key aspect of the*

*manuscript that is lacking, which is important given it is central to the re-interpretation of the results of Painter et al. 2013. In this context, it may be worth pointing out that in Figure 2 of Painter et al. 2013, the lines for Fiescherhorn and Colle Gnifetti ice cores seem to have been mislabeled as the opposite core.*

We added a section focusing on the comparison to the EC record from Thevenon 2009. Thanks for pointing out their mislabeled Figure. We too were very surprised this mistake has not been noticed by their reviewers and editor during the peer-review process and added a note of caution so readers do not get confused.

*Editorial Comments*
*p6 ln17 – "both, our" – remove comma.*
*p15 ln4 – Data availability details are missing*
*Supplementary Fig. S5: Is the correlation shown in panel c a Pearson's coefficient? If so, please state.*

Changed as suggested. Data availability details will be added upon acceptance of the manuscript for publication.

References (not already cited in the manuscript)
Goosse, H., Barriat, P.-Y., Dalaiden, Q., Klein, F., Marzeion, B., Maussion, F., Pelucchi, P., and Vlug, A.: Testing the consistency between changes in simulated climate and Alpine glacier length over the past millennium, Climate of the Past Discussions, doi: 10.5194/cp-2018-48, 2018. 1-27, 2018.

Zekollari, H., Fürst, J. J., and Huybrechts, P.: Modelling the evolution of Vadret da Morteratsch, Switzerland, since the Little Ice Age and into the future, Journal of Glaciology, 60, 1155-1168, 2017.

Preunkert, S., Wagenbach, D., and Legrand, M.: A seasonally resolved alpine ice core record of nitrate: Comparison with anthropogenic inventories and estimation of preindustrial emissions of NO in Europe, 
[revised manuscript text omitted]

---

## Author Response (AR3)

**Reply to Editor**

We would like to thank the editor for her detailed suggestions and her constructive commentary that has helped us improve the manuscript. The editor's comments are printed in **green** font, with our responses in **black** and our textual changes marked in **_italic black font_**. We here resubmit a revised version with tracked changes for your consideration.

-------------------------------Editor (Remarks to the Author):-------------------------------------------------
Dear Dr. Sigl,
Thank you for your consideration of the 3rd reviewer's comments and for your revisions. I read through your replies and revisions carefully. I think that you mostly (but not entirely) have addressed the reviewers' comments. Your additions to the manuscript were a bit frustrating to read as they contained many overly long, run-on, grammatically incorrect sentences. I have noted this specifically below. I've also suggested some wording changes that I think will better respond to the reviewers' concerns and make the manuscript more clear. Please submit your revisions as tracked changes to me after carefully reading through and editing your writing. Page and line numbers refer to the tracked changes version of your revisions.
Regards,
Becky Alexander

We are pleased to see that we have (almost entirely) addressed the minor revisions suggested by reviewer 3. We can assure you that writing in a foreign language is not an easy task and we apologize if our sentences were sometimes too long and not always grammatically correct. We tried to improve on this in the current revision and addressed the remaining concerns in response to your report.

**Title:**

The title is too long. It would be better to simply say "19th century retreat in the Alps preceded the emergence of industrial black carbon deposition on high-alpine glaciers" as that is what your paper clearly shows.

Changed as suggested: _"19th century retreat in the Alps preceded the emergence of industrial black carbon deposition on high-alpine glaciers"_

**Abstract:**

Page 1 Line 26: What does "well-replicated" mean? This is vague. Did you do replicate analysis of samples or is based on your stacked data set? I think it's the latter, but it would be better to choose more clear wording.

We actually meant both 1) the analytical long-term repeatability, i.e. the agreement of measurements on parallel replicate core sections analyzed in the same laboratory at different times to demonstrate that the results are not an artefact of the measurement procedures and 2) the result reproducibility, i.e. the ability to independently achieve similar conclusions (e.g., between CG03 rBC and FH02 EC) with different sampling techniques, analytical methods and data processing methods.

We removed "_well replicated_" here since we discuss reproducibility already in the according sections of the manuscript.

Page 1 lines 32 to end of abstract: I suggest ending the abstract as follows beginning on line 32: "…than 80% of their total 19th century length reduction, casting doubt on a leading role for anthropogenic BC emissions in terminating the Little Ice Age. Attribution of glacial retreat requires expansion of spatial network and sampling density of high alpine ice cores to balance potential biasing effects arising from transport, deposition and snow conservation in individual ice core records." I don't think the reviewers will be happy with the sentence beginning on line 32 ("Industrial BC emissions…") as you have not done any forcing calculations as the sentence may imply. The sentence beginning on line 33 ("BC records…") comes out of nowhere and doesn't make sense here.

Changed as suggested: *"…than 80% of their total 19th century length reduction, casting doubt on a leading role for anthropogenic BC emissions in terminating the Little Ice Age. Attribution of glacial retreat requires expansion of spatial network and sampling density of high alpine ice cores to balance potential biasing effects arising from transport, deposition and snow conservation in individual ice-core records."*

However, we can't follow your argument why radiative forcing calculations should be necessary in support of our previous statement. We had already restricted our previous statement to "**primary** *forcing for the* **initial rapid deglaciation**", leaving open the possibility that rBC may have contributed to some extent to the overall 19th century glacier retreat. Following the IPCC AR4, "*radiative forcing is a measure of the influence a factor has in altering the balance of incoming and outgoing energy in the Earth-atmosphere system and is an index of the importance of the factor as a potential climate change mechanism. In this report radiative forcing values are for changes* **relative to pre-industrial conditions** *defined at 1750*". We find -- based on all ice-core rBC records that we discussed in our study -- that industrial rBC deposition between 1850 and 1875 AD was not distinguishably different from the pre-industrial background. Therefore, the BC in snow radiative forcing before 1875 AD was by definition zero.

**Introduction:**
Page 2 Line 11-12: Change to "…because it absorbs solar radiation even at very low concentrations…"
Changed as suggested.

Page 3 lines 1-4: This is an example of a long run-on sentence that is difficult to make sense of. It needs to be split into two sentences. Perhaps "Together with mineral dust and other absorbing organic aerosols, BC deposited on snow and ice can lead to increased melt rates and changes in melt onset due to reductions in surface albedo. These effects are further enhanced by subsequent snow albedo feedbacks such as an increase in the water content and surface accumulation of impurities."
Changed as suggested.

Page 3 line 5: Should read "…best estimate for industrial era global forcing of BC is…"
Changed as suggested.

Page 4: This new section is difficult to read. The first sentence is long and grammatically incorrect. I cannot tell exactly what you are trying to say. I'll leave it up to you and your co-authors to split this up into two or more sentences and check it for grammar and correctness.

We have split this section into three sentences and shortened it slightly.

The next section beginning with "The snow-albedo feedback" on line 18 also contains long, run-on sentences and is very disorganized making it difficult to read. You are casting doubt on the Painter et al study for three reasons: 1. A low resolution data set, 2. Measurements were performed using questionable analytical methods and 3. Large dating uncertainties. Perhaps you could start it like this: *"The snow-albedo feedback hypothesis formulated by Painter et al was a first effort to attempt this but was limited by the available BC data at that time[here provide references for the data that was used in this study]. The available data were of relatively low time-resolution [reference specific data set(s)] and the dating was uncertain [reference specific data set(s)]."* The phrase "compiled from very first applications using different analytical methods" doesn't really say anything. Just because they are the "first" measurements or "different" (different from what?) doesn't mean they aren't any good. You need to be specific on what you think is wrong with the analytical technique and provide appropriate references to support this. "Some of these methods" – which methods? Both Jenk et al and Thevenon et al or just one of them? At this point the reader doesn't know how many data sets you are referring to and if they all suffer from the same issues. How do you know that a method(s) did not deliver reproducible results? Is there a reference for this? Personal communication? You need to support such a claim. Perhaps you should follow this *"The snow-albedo feedback hypothesis formulated by Painter et al was a first effort to attempt this but was limited by the available BC data at that time[here provide references for the data that was used in this study]. The available data were of relatively low time-resolution [reference specific data set(s)] and the dating was uncertain [reference specific data set(s)]."* with *"Here, we set out to re-evaluate the timing of industrial BC deposition in ice cores from the alps by using a new, more accurately dated record of light absorbing aerosols at much higher time resolution (sub-annual). In addition, we measure distinctive tracers of anthropogenic pollution (list them here) and compare all records with the most highly resolved history of glacier length changes of four glaciers in the Western Alps currently available."* This removes some of the details (which are unclear for reasons stated above), and I suggest moving this more detailed discussion into section 3.2 I also suggest removing the last sentence of this section beginning with "We determine…"

We have reformulated this section following your advice.

*"Transient changes in external natural (e.g. volcanic eruptions) and anthropogenic climate forcing (e.g. greenhouse gases, tropospheric aerosols) occurred during the emergence of the industrialization in Europe (Eyring et al. 2016; Jungclaus et al., 2017). To isolate the often-complex relationships between glacier fluctuations and meteorological forcing and to identify the mechanisms responsible for glacier retreat in the second half of the 19th century requires comprehensive modelling efforts (e.g., Lüthi 2014; Zekollari 2017, Goosse et al., 2018). Underpinning such efforts, accurate and precise delineation of external forcing (e.g., volcanic eruptions), potential feedbacks (e.g., BC deposition on snow) and cryosphere changes (e.g., variations in glacier front positions) is critically important.*

*The snow-albedo feedback hypothesis formulated by Painter et al. (2013) was a first effort to attempt this but was limited predominantly by the available BC data at that time (Thevenon et al., 2009; Jenk et al., 2006). The available data were of relatively low time-resolution (Thevenon et al., 2009; Jenk et al., 2006), and the dating of the Colle Gnifetti ice core was at that time not based on annual-layer dating constrained by historic age markers (Jenk et al., 2009; see Supplementary Fig. S 1, Supplementary Table S1) and therefore was highly*

*uncertain in Thevenon et al. (2009). Measurements of EC maybe subject to artifacts related to losses during filtration, interferences from mineral dust or the pyrolysis of organic compounds (see Lack et al., 2014; Lim et al., 2014 for details). Large sample size requirements (0.2-1 kg) for EC quantification with traditional thermal techniques, however, made impossible to analyze replicate core sections in order to demonstrate the repeatability of the results (Thevenon et al., 2009; Jenk et al., 2006).*

*Here, we set out to re-evaluate the timing of industrial BC deposition in ice cores from the Alps by using a new, more accurately dated record of rBC at much higher time resolution (sub-annual). In addition, we measure distinctive tracers of anthropogenic pollution (bismuth, sulphate, lead, ammonium) and compare all records with the most highly resolved history of glacier length changes of four glaciers in the Western Alps currently available (Nussbaumer & Zumbühl 2012)."*

**Section 3.2:**

Page 10 line 20: What do you mean by "non-equidistant samples"? "not-validated method" is not supported – see also my previous comment from the introduction.
Non-equidistant samples do not have the same time-resolution. In Thevenon et al. (2009), samples taken in the youngest part have a nominal time resolution of about 2 years, whereas samples taken from the older part can contain >50 years. This varying resolution makes it impossible to interpret long term trends. Since the highest EC concentrations from Thevenon et al. (2009) when averaged over 30 years are found in the pre-industrial era (around 1550 AD) and not in the 20[th] century (in contrast to any other ice core from Greenland and Europe analyzed so far) we have strong doubts that these results would have been repeatable (but no replication analyses had been performed).
We clarified this point by writing *"In contrast, the low-resolution EC record of Thevenon et al., (2009) -- characterized by varying time resolution and without demonstrated repeatability of the results -- shows a markedly different variation in time."*

Page 10 line 23: remove comma after "14 years"
Done

Page 10 line 26: Should read "reach comparable levels during the peak"
Changed as suggested.

Page 10 line 27: Replace "assume" with "speculate"
Done

**Discussion:**

Page 13 lines 8 – 9: Should read "82% [52%] of the glacier length reductions had already occurred at the best [earliest] estimated time of emergence of industrial BC deposition." I removed the end of this sentence because you can't say it played no role later on.
Changed as suggested.

Page 13 line 12: "1900 AD AND also by"
Done

Page 13 line 14: remove comma after 1875 AD

Done

Page 13 line 19: insert comma after Sun et al reference.
Done

Page 13 lines 20-24: Should read: "However, these species are measured by the method applied for the EC record from FH02 which shows very good agreement with the rBC record from Colle Gnifetti (Section 3.2). This suggest that factors other than changes in surface snow albedo, such as temperature and seasonal precipitation distribution (Steiner, Zumbuhl), may have dominated mass balance and glacier length variability of these European glaciers until at least 1875 AD."
Changed to: *"However, these compounds are measured by the method applied for the EC record from FH02 which shows very good agreement with the rBC record from Colle Gnifetti (Section 3.2). This suggest that factors other than changes in surface snow albedo, such as temperature and seasonal precipitation distribution (Steiner et al., 2008; Zumbühl et al., 2008), may have dominated mass balance and glacier length variability of these European glaciers until at least 1875 AD."*

Page 13 line 26: remove comma after variability
Done

Page 14 line 10: change "can thus be understood as" to "may be"
Done

Page 14 line 14: remove "to some extent".
Done

Page 14 line 19-21: Should read "…will require reconciliation of early instrumental and proxy climate data [references] and the use of models to decompose…"
Changed as suggested.

Page 14 lines 25-29: Remove the last 3 sentences. This belongs in the conclusions.
Removed as suggested.

**Conclusions:**
Page 15 line 28: replace "enhancing" with "reductions in"
Corrected as suggested.

Page 15 line 32-33: Should read "species co-analyzed enabled BC source attribution from industrial and biomass burning emissions."
Changed as suggested.

Page 16 line 7: Should read "We hypothesize that glacier length changes throughout the past 2,000 years have been…"
Changed as suggested.

Page 16 line 15: Remove last sentence of this paragraph.
Done

Page 16 line 16: Replace "computer" with "model"
Done

Page 16 line 18: Should read "the past few decades"
Changed as suggested.

Page 16 line 20: Should read "model evaluation" (remove the "s")
Done.

Page 16 lines 21-22: Should read "Here we present the first continuous…"
Changed as suggested.

[revised manuscript text omitted]